# N-cadherin crosstalk with integrin weakens the molecular clutch in response to surface viscosity

Eva Barcelona-Estaje [1], Mariana A. G. Oliva [1], Finlay Cunniffe[1], Aleixandre Rodrigo-Navarro [1], Paul Genever[2], Matthew J. Dalby [1], Pere Roca-Cusachs [3,4] ✉, Marco Cantini [1] ✉ & Manuel Salmeron-Sanchez [1,3,5] ✉

Mesenchymal stem cells (MSCs) interact with their surroundings via integrins, which link to the actin cytoskeleton and translate physical cues into biochemical signals through mechanotransduction. N-cadherins enable cell-cell communication and are also linked to the cytoskeleton. This crosstalk between integrins and cadherins modulates MSC mechanotransduction and fate. Here we show the role of this crosstalk in the mechanosensing of viscosity using supported lipid bilayers as substrates of varying viscosity. We functionalize these lipid bilayers with adhesion peptides for integrins (RGD) and N-cadherins (HAVDI), to demonstrate that integrins and cadherins compete for the actin cytoskeleton, leading to an altered MSC mechanosensing response. This response is characterised by a weaker integrin adhesion to the environment when cadherin ligation occurs. We model this competition via a modified molecular clutch model, which drives the integrin/cadherin crosstalk in response to surface viscosity, ultimately controlling MSC lineage commitment.

Cells dynamically interact with the extracellular matrix (ECM) through transmembrane receptors called integrins, which connect the ECM and the actin cytoskeleton and enable the transmission of physical cues from the ECM[1]. Native ECMs are viscoelastic. However, synthetic materials designed to study cell mechanotransduction typically focus on the modulation of elastic stiffness and how cells react to this property[2–4]. Only recently have biomaterial designs focused on the development of viscoelastic materials to better mimic native ECMs and understand cellular behaviour[5]. These mechanically dynamic and dissipative environments are known to modulate important processes, such as cell adhesion, migration, proliferation, and differentiation [6,7].

Although the ECM presents with both elastic and viscous properties, it is important to uncouple how cells react to each of these cues.

In this work, we investigate how viscosity affects stem cell mechanotransduction (particularly in human mesenchymal stem cells, hMSCs), as this process ultimately affects stem cell differentiation[8]. Viscosity defines the range of motion, or mobility, of the molecules in a material, and several earlier studies have investigated the effects of viscosity on cell adhesion[9], spreading[10] and fate[11]. In particular, we have previously investigated integrin-mediated mechanisms of cell response to surface viscosity. To do so, we functionalized supported lipid bilayers (SLBs) of different viscosity with RGD and explained the cell mechanotransduction of viscosity using the framework of the molecular clutch[12]. This clutch engages when the stiffness of the ECM is sufficiently high to expose binding sites in mechanosensitive proteins. We have also used this measure to interpret cell response to viscosity[12]. The molecular clutch framework models the interactions between

[1]Centre for the Cellular Microenvironment, Advanced Research Centre, University of Glasgow, Glasgow, UK. [2]Department of Biology, University of York, York, UK. [3]Institute for Bioengineering of Catalonia (IBEC), the Barcelona Institute of Technology (BIST), Barcelona, Spain. [4]University of Barcelona, Barcelona, Spain. [5]Institució Catalana de Recerca i Estudis Avançats (ICREA), Barcelona, Spain. ✉e-mail: proca@ibecbarcelona.eu; Marco.Cantini@glasgow.ac.uk; Manuel.Salmeron-Sanchez@glasgow.ac.uk

actomyosin cytoskeleton force generation and dynamics, cell migration dynamics, and mechanotransduction. It explains the dynamic nature of the cytoskeleton-ECM linkage and its relationship to cell mechanosensing[13]. The continuous flow of actin towards the cell nucleus, as created by actin polymerization and myosin contractility, is countered by the resistance of the ECM, as actin filaments are coupled to it through focal adhesions (FAs) and integrins. In a purely viscous substrate, low viscosity leads to low ECM resistance, promoting faster actin flows. At high viscosities, higher ECM resistance leads to slower actin flows. This mechanosensing of viscosity is translated into biochemical signals, which ultimately influence gene expression[12].

Within the cellular microenvironment, cells are in contact with the ECM and also interact with other cells via cadherins. These transmembrane adhesion proteins play a crucial role in development, tissue homoeostasis and wound healing[14]. Like integrins, cadherins are connected to the actin cytoskeleton, meaning that their adhesion machinery is also mechanosensitive[15]. Indeed, cadherin adhesion clusters, or adherens junctions (AJs), have an analogous structure to FAs. In the case of FAs, integrins are linked to talin, which unfolds under force[16]. Talin unfolding triggers vinculin adhesion, reinforcing the FA complex, and the subsequent attachment of the actin cytoskeleton[16–19]. In the case of cadherins, α-catenin unfolds under force, similarly to talin[20], exposing a vinculin adhesion site[21]. Both α-catenin and vinculin can then bind to actin filaments, producing cadherin-based mechanotransduction[22–24]. Both integrin and cadherin-mediated mechanotransduction pathways converge at multiple points, creating what has been referred to in the literature as 'adhesive crosstalk'[25,26]. To understand the role of this adhesive crosstalk in cell mechanotransduction, several biomaterials that include both integrins and cadherin adhesive domains have been developed[27–29]. However, the role of cadherins in the mechanosensing of viscoelasticity, and viscosity in particular, remains unclear.

This crosstalk between integrins and cadherins is also physiologically relevant in processes such as collective cell migration, that underpins development, cancer metastasis and regeneration[5,30,31]. The role of the integrin-cadherin crosstalk in cell migration has been investigated as a response to substrate stiffness[32]. Yet, the ECM is viscoelastic in nature and there is increasing interest in understanding cell response to it. Apart from a few seminal papers in the field, some excellent reviews have been published in the last few years[7]. However, while significant amount of work has been done to understand cell response to elasticity, including the role of integrins and cadherins[27], the role of viscosity is not understood yet. Our work lies in the context of understanding the role of the viscous part of viscoelasticity and how this impacts integrins and cadherins. Integrins are key to interact with the ECM—of viscoelastic nature—whereas cadherins link cells and their role is modulated by the viscosity of the cell membrane. It has been previously shown that a stiffness-dependent competition occurs between integrin and cadherin mechanosensitive pathways, which reduces YAP translocation to the nucleus when both integrins and cadherins are engaged[27,29,33]. In this study, we demonstrate that adhesive crosstalk also influences the mechanosensing of viscosity.

Crucially, due to the linkage of both FAs and AJs to the actin cytoskeleton by analogous molecules[20], we hypothesise that the competition between these complexes occurs at the point of binding to the actin filaments. To prove this, we used SLBs of different viscosity, DOPC and DPPC SLBs, which have different states—fluid and gel phases respectively—in cell culture conditions. Together with a non-mobile glass control, these SLBs were functionalized with an adhesion peptide from integrins (RGD) alone or with a mixture of RGD and varying concentrations of HAVDI, an adhesion peptide from N-cadherins. Using this purely viscous system, we demonstrate the existence of a viscosity-dependent competition for actin filaments in adhesive crosstalk in hMSCs, which we have modelled using a modified molecular clutch model[12].

## Results

### N-cadherin ligation affects cell morphology in a viscosity-dependent manner

We fabricated DOPC and DPPC SLBs using the vesicle deposition method as previously described[12]; DOPC and DPPC SLBs were chosen due to their different viscosities in cell culture conditions (Fig. 1a); these have been previously estimated to be $1 \times 10^{-6}$ Pa·s·m for DOPC and $1 \times 10^{-4}$ for DPPC. SLBs were functionalised with NeutrAvidin (NA) only or with RGD at a 0.2 % mol concentration. We then confirmed the specific adhesion of hMSCs to the RGD adhesion ligand. Our results show (Fig. 1b–d) that the number of attached hMSCs increased significantly only on surfaces functionalised with RGD (Fig. 1c, d), proving the non-fouling nature of the bilayers in the absence of adhesive ligands.

Previous work has shown that viscosity has a significant effect on cellular response[9,10,12,34,35]. Indeed, low viscosity lipid bilayers have been demonstrated to support smaller cells with a more rounded morphology[36,37]. Here, we studied hMSC behaviour on surfaces functionalised with different concentrations of RGD (Supplementary Fig. 1). We observed that cell spreading increased significantly with increasing viscosity, independently of RGD levels, producing the highest levels of cell spreading in the non-mobile glass control. This response has been previously observed with another adherent cell type[12]. We also tagged the bilayers with a fluorescent marker to confirm bilayer stability during culture (Supplementary Fig. 2i); while we observed some reorganization of the bilayers by the cells, this is expected as mechanical remodelling at the cell-material interface is a known phenomenon related to substrate compatibility[38–40]. Despite this reorganisation, cell response remains governed by the initial interactions and by the varying degree of viscosity of the substrate, as observed in Fig. 1a–d and in Supplementary Fig. 1.

We next studied the role of adhesive crosstalk by functionalizing bilayers with 0.2 % mol RGD and with 0.02 or 2% mol HAVDI (referred to as "low HAVDI" or "high HAVDI", respectively). We tagged the adhesion peptides with fluorescent markers to visualize and quantify the functionalized bilayers (Supplementary Fig. 2). Although some studies in hydrogel systems have reported that there is no difference in cell area in the presence of N-cadherin ligands[27,29], here we observed a decrease in cell spreading on the bilayers when they were also functionalised with HAVDI (Fig. 1e–g). This observed decrease in cell area was abrogated when N-cadherin was blocked with an antibody, confirming specific cadherin adhesion to the HAVDI ligands (Fig. 1e–g). We also observed a higher intensity of N-cadherin expression, as assayed by immunofluorescence, in cells seeded on surfaces functionalised with RGD and HAVDI (Supplementary Fig. 3).

When comparing RGD density in our work to the above-mentioned studies, we found that the theoretical distance among RGD ligands is 44 nm in this study (calculated as $1/\sqrt{n}$, where n is the particle density)[12], compared to 150[29] and 74 nm[27] in the other studies. While these values are within the same order of magnitude, lower distances between RGD ligands could cause a stronger cell attachment, which is then disrupted by the HAVDI peptides, hence producing a decrease in cell area. Indeed, when lower concentrations (0.02% mol) of RGD were tested (i.e. higher RGD spacing), changes in the size of hMSC area due to HAVDI were observed only for DPPC and not for DOPC (Supplementary Fig. 4a–c). We also studied integrin $\alpha_v$ because it has been previously reported to undergo clustering after interacting with RGD, resulting in the recruitment of additional adhesion molecules, such as talin, paxillin or FAK[41] (Supplementary Fig. 4d–j). We observed no effect of HAVDI on integrin clustering at the lower 0.02% RGD concentration compared to 0.2% mol RGD.

### N-cadherin ligation affects hMSC mechanosensing and differentiation

After investigating how N-cadherin adhesion influences the initial cell response to viscosity, we next analysed YAP localization. Previous

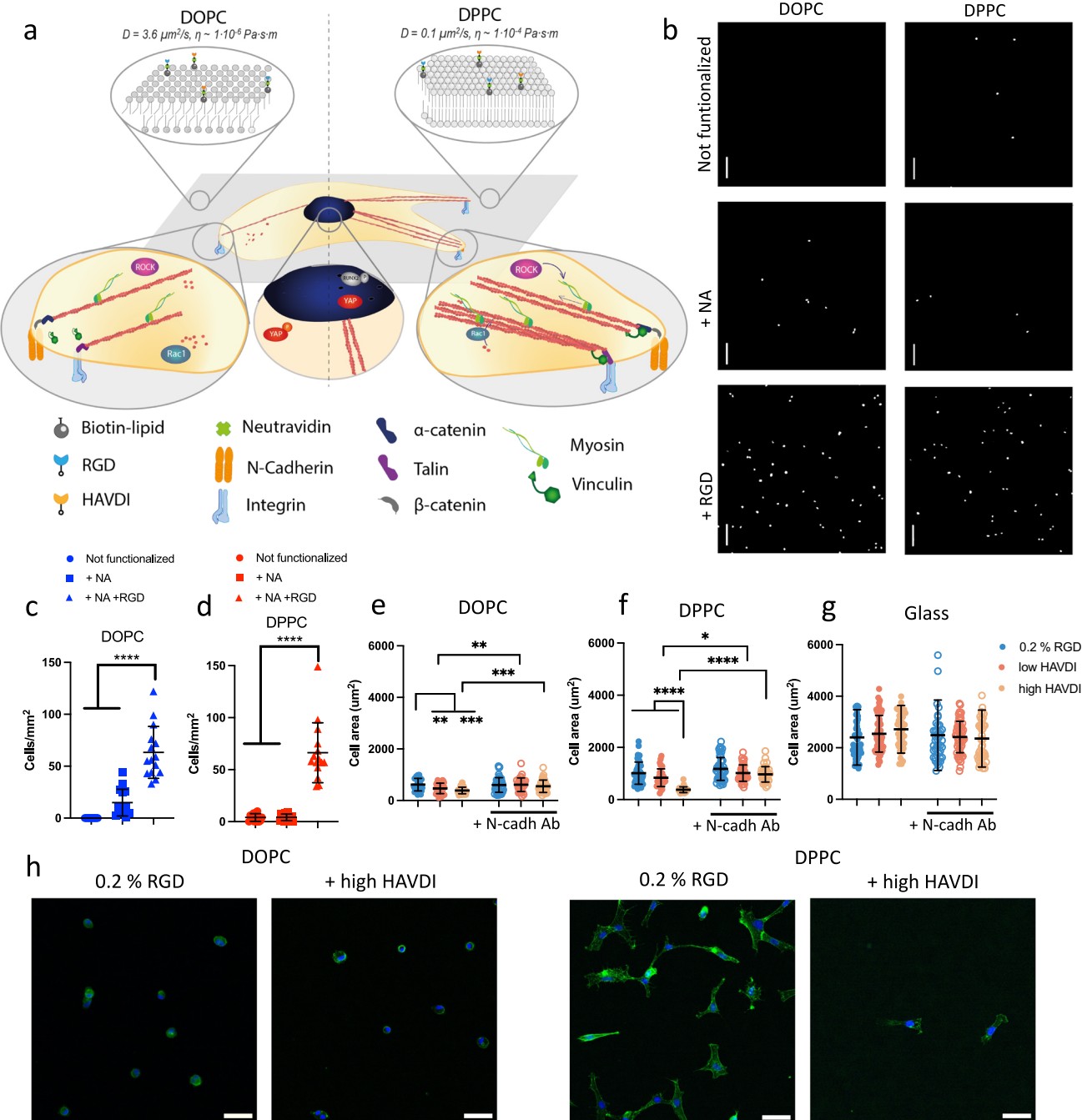

**Fig. 1 | Supported lipid bilayers functionalized with integrin and N-cadherin adhesive peptides as a platform to study adhesive crosstalk in hMSC adhesion and mechanosensing. a** Schematic representation of hMSCs seeded on DOPC (left) and DPPC (right) supported lipid bilayers functionalized with RGD and HAVDI peptides. Viscosity values taken from ref. 12. **b** Representative images of DAPI stained hMSCs seeded on bilayers for 24 h without functionalization, with only NA and with NA and RGD. Cell numbers quantification of hMSCs seeded for 24 h (scale bar = 50 μm) on (**c**) DOPC and (**d**) DPPC SLBs without functionalization, with only NA or with RGD (*n* = 15 in all conditions). On non-functionalised DOPC, no cells were found on any sample. **e**–**g** hMSCs area after 24 h of cell culture on DOPC, DPPC and

glass, respectively, with 0.2% mol RGD with low or high HAVDI without treatment or with N-cadherin blocked (+N-cadh Ab conditions) (*n* (graph (**e**)) from left to right 36, 38, 54, 45, 30, 36. *n* (graph (**f**)) from left to right 47, 34, 30, 47, 45, 37. *n* (graph (**g**)) from left to right 53, 59, 49, 45, 59, 50). All *n* represents cells. **h**) Representative images of hMSCs seeded on DOPC and DPPC bilayers functionalized with only RGD or RGD and HAVDI (Actin: green, DAPI:blue). All data are presented as mean values ± SD. (Scale bars = 50 μm) Statistical significance was determined using D'Agostino Pearson normality test, followed by a Kruskal–Wallis multiple-comparison test. All data are presented as mean values ± SD.*$P \leq 0.05$, **$P \leq 0.01$, ***$P \leq 0.001$, ****$P \leq 0.0001$.

studies have shown that YAP translocates to the nucleus in cells in environments with higher stiffness[42] or viscosity[12], due to the enlargement of the nuclear pores caused by an increase in the force transmitted to the nucleus[43]. In the case of HAVDI ligation, previous studies have reported that at intermediate stiffnesses (10–15 kPa), YAP translocation to the nucleus is decreased, while in softer or

stiffer substrates, HAVDI does not modify YAP location, remaining in the cytoplasm or nucleus, respectively[27,29,33]. Here, we studied YAP behaviour after HAVDI ligation in viscous surfaces (Fig. 2a). After 24 h of cell culture, YAP translocated to the nucleus of hMSCs on DPPC and glass. We observed the highest levels of nuclear YAP in cells seeded on glass controls (Fig. 2b, Supplementary Fig. 5). By contrast,

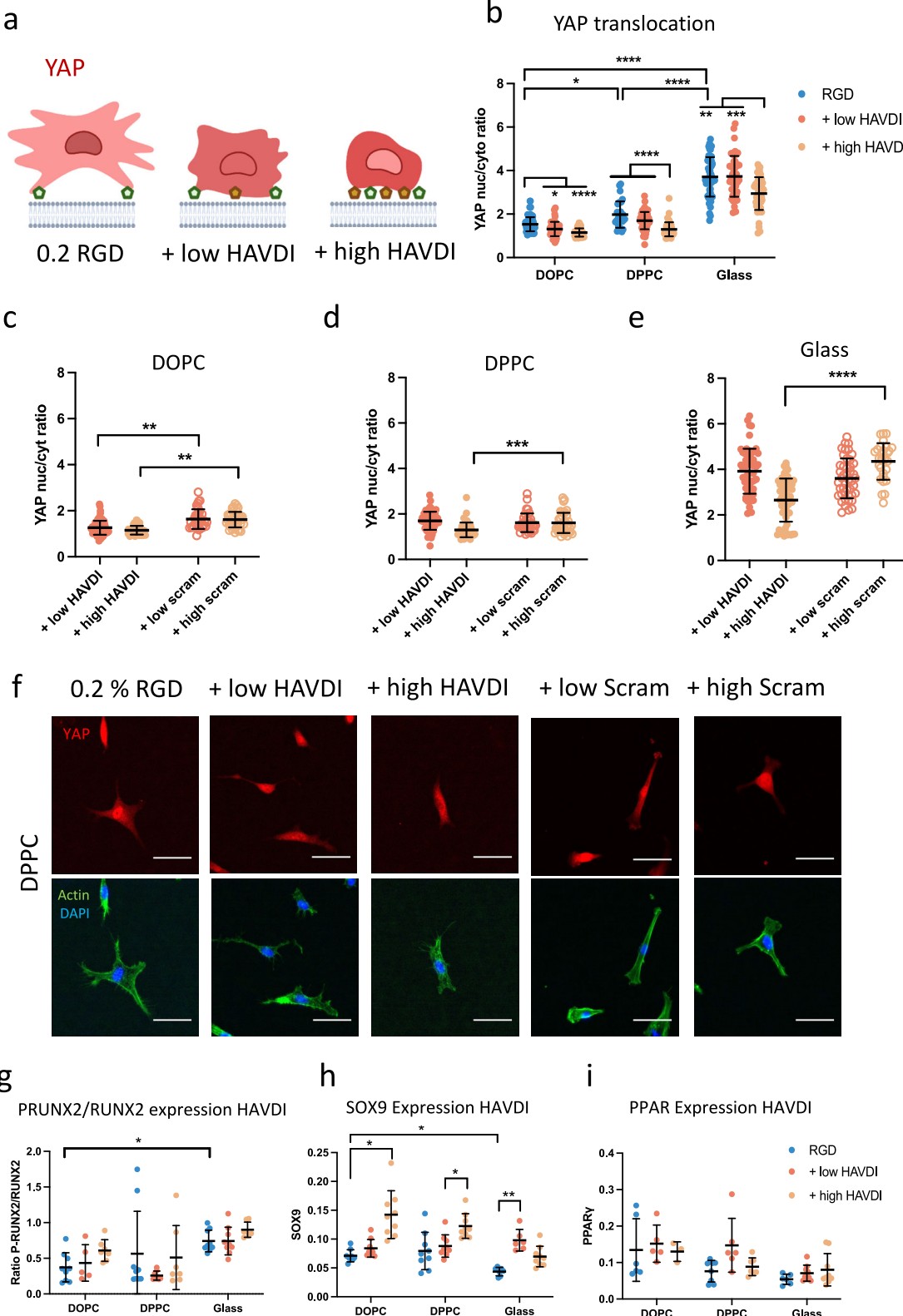

YAP nuclear levels decreased when surfaces were functionalized with low or high HAVDI, independently of viscosity (Fig. 2b).

We next used a scrambled peptide that cannot be recognized by N-cadherin to confirm that differences in YAP translocation were caused by N-cadherin signalling and not merely by the presence of a peptide impeding integrin ligation. Our results show (Fig. 2c–e) that hMSCs seeded on surfaces that contained the scrambled peptide maintained high nuclear levels of YAP compared to those seeded on surfaces that contained HAVDI, confirming that the decrease in nuclear YAP is produced by an N-cadherin-mediated alteration in integrin mechanotransduction. In elastic substrates, reductions in YAP translocation to the nucleus have been previously attributed to competition between the N-cadherin and Integrin signalling pathways in the regulation of hMSC contractility. It has also been

**Fig. 2 | HAVDI ligation reduces YAP nuclear translocation independently of viscosity. a** Schematic representation of YAP translocation to the nucleus of hMSCs on surfaces with RGD and RGD with low and high HAVDI. Image made with BioRender. **b** Quantification of YAP translocation of hMSCs seeded on SLBs and glass functionalized with RGD and RGD + low and high HAVDI (n from left to right = 36, 36, 32, 15, 34, 21, 20, 21, 16). All n represent cells. Statistical significance was determined using D'Agostino Pearson normality test, followed by a mixed-effects analysis. Quantification of YAP translocation to the nucleus of hMSCs seeded on (**c**) DOPC (from left to right n = 75, 59, 37, 38), (**d**) DPPC (from left to right n = 61, 48, 34, 38), and (**e**) glass (from left to right n = 61, 58, 45, 29) functionalized with low and high HAVDI or scrambled HAVDI peptide; statistical differences in (**c**–**e**) are shown only between HAVDI and scrambled HAVDI conditions. Statistical significance was determined using D'Agostino Pearson normality test, followed by a Kruskal−Wallis multiple-comparison test. **f** Representative images of hMSCs seeded on DPPC functionalized with RGD, RGD + low and high HAVDI or RGD + low and high scrambled HAVDI (scale bar = 50 μm). **g–i** show early differentiation marker expression measured by In-Cell Western of hMSCs for osteogenesis (7 days) (from left to right n = 9, 5, 7, 9, 8, 7, 9, 9, 7), chondrogenesis (from left to right n = 9, 9, 9, 9, 9, 7, 6, 9) and adipogenesis (5 days) (from left to right n = 6, 5, 3, 8, 6, 7, 6, 9, 9) respectively. In graphs (**g**–**i**) n represent areas in the in-cell well. All statistical differences for (**g**–**i**) can be seen in Supplementary Tables 2–4. Statistical significance was determined using D'Agostino Pearson normality test, followed by followed by a mixed effects analysis. All data are presented as mean values ± SD. *$P \le 0.05$, **$P \le 0.01$, ***$P \le 0.001$, ****$P \le 0.0001$.

previously demonstrated that HAVDI inhibits Rac1 activity, decreasing myosin IIA (MIIA) location in focal adhesions and cofilin phosphorylation[27,29]. In this study, we investigated MIIA location in focal adhesions and found that adding HAVDI decreases MIIA levels in focal adhesions compared to RGD functionalised DPPC surfaces (Supplementary Fig. 6). More studies are needed to conclude that the same mechanisms are used in the mechanosensing of viscosity.

hMSCs can differentiate towards bone, cartilage, and fat tissue under the influence of matrix stiffness sensed by integrins[44]. hMSC mechanosensing thus modulates the hMSC transcriptional response, impacting their differentiation. However, integrins are not the only ligand that can modulate lineage commitment; N-cadherin engagement has also been shown to promote chondrogenesis[28]. Here, we studied early differentiation markers in hMSCs in the presence of RGD or HAVDI peptides, specifically Phospho (P)-RUNX2 for osteogenesis, SOX9 for chondrogenesis, and PPARγ for adipogenesis (Fig. 2g-1)[45]. We observed an increase in P-RUNX2 expression at increasing viscosity in the presence of RGD (Fig. 2g and Supplementary Fig. 7a), which was matched by increased cell area and increased YAP nuclear translocation (Figs. 1e, f and 2b). These results support previously reported findings, in which YAP and TAZ were demonstrated to act as transcriptional regulators of RUNX2[46].

In the presence of HAVDI peptides, we observed a trend of increased P-RUNX2 expression at increasing concentrations of HAVDI, albeit the differences were not significant (Fig. 2g). With SOX9, we observed that expression decreased with increasing viscosity (Fig. 2h, Supplementary Fig 7b). Interestingly, at increasing concentrations of HAVDI, SOX9 expression increased independently of viscosity. These results agree with previous studies, which have reported increased chondrogenesis after N-cadherin adhesion[28,47], and that the blocking of N-cadherin postpones early cartilage differentiation[48]. On glass, there was instead a reduction in SOX9 expression with high HAVDI, which might be related to an increase in osteogenesis[49]. PPARγ showed higher expression levels at lower viscosities in the presence of RGD (Fig. 2i, Supplementary Fig. 7c), a trend opposite to that observed for osteogenesis. While it has previously been reported that softer materials promote adipogenesis[50], here we show that viscosity also plays a role in adipogenic differentiation. We observed higher PPARγ expression on surfaces with a lower force of adhesion, less engagement of the molecular clutch, and where YAP was predominantly cytoplasmic, in agreement with the literature[51]. N-cadherin signalling, however, was not found to influence adipogenesis (Fig. 2i).

## N-cadherin ligation disrupts the engagement of the molecular clutch

The mechanoresponse of stem cells to substrate viscosity, which eventually determines their fate as evidenced in Fig. 2, can be interpreted through a molecular clutch model (Fig. 3a, b). In agreement with previous work done with a murine cell line[12], we observed that higher viscosities (DPPC and immobile glass) produced an increase in hMSCs FA intensity and length, as measured by P-FAK

(Fig. 3c, d and Supplementary Fig 8b) and vinculin immunostaining, respectively (Fig. 3e and Supplementary Fig 8b). We noted an increase in the intensity of P-FAK with viscosity, indicating that more viscous surfaces produce an increase in the force of adhesion[52]. This behaviour is predicted by a computational clutch model in which the RGD ligands are bound to a viscous dashpot that represents the substrate's viscous behaviour (Fig. 3f, RGD only)[12]. On DOPC, the clutch is not engaged, leading to the formation of small adhesions, while at increasing viscosity (on DPPC), talin unfolding leads to the engagement of the clutch and to reinforcement of the adhesions (Fig. 3d−f, RGD only).

We also observed a viscosity-dependent behaviour when the surfaces included low or high HAVDI (Fig. 3a−e). Cells on DOPC did not show any change in FA formation compared to RGD-only surfaces (as shown by both vinculin and P-FAK staining). By contrast, hMSCs showed a decrease in the length and intensity of FAs when seeded on more viscous surfaces (DPPC) or on glass controls containing HAVDI. These results are in accordance with how the adhesive crosstalk affects FA formation on hydrogels of varying stiffness in terms of FA length[27,29].

Since N-cadherin and integrin mechanosensing pathways converge at multiple points, sharing proteins such as vinculin, FAK or Rho GTPases (which remodel the actin cytoskeleton[25]), it is likely that N-cadherin ligation would disrupt the engagement of the molecular clutch[33]. Here, we hypothesise that the clutch is affected by the competition for actin filaments between FAs and AJs. Indeed, both structures are connected to F-actin, and α-catenin is a homologous protein to vinculin[20]. In this model, talin-vinculin and α-catenin-vinculin would compete with each other to bind to actin, given the limited availability of actin filaments (Fig. 3a, b). N-cadherin homophilic interactions are weaker than integrin-ECM interactions; for this reason, as the HAVDI concentration increases, a greater number of weaker bonds are created. This produces the observed smaller FAs (Fig. 3c−e, see Supplementary Fig 8b for representative images of P-FAK and vinculin staining) and lower YAP nuclear translocation (Fig. 2b). To model the adhesive crosstalk in response to viscosity, we modified the computational clutch model described previously[12] to account for the competition of RGD- and HAVDI-based adhesions for actin binding (Supplementary Note 1).

In the model, when HAVDI concentrations are higher, actin-ECM links (clutches) are increasingly mediated by cadherins rather than by integrins. Since cadherin-cadherin bonds have a lower resistance to force than integrin-ECM bonds do, and since they do not mediate FA reinforcement, the increased formation of cadherin-cadherin bonds leads to a decreased response to viscosity. Indeed, this model predicts that the functionalization of viscous surfaces with HAVDI produces a significant decrease in the size and number of FAs only at higher viscosity (Fig. 3d, e and Supplementary Fig 8a, respectively). This prediction is thus supported by our experimental results. When low-viscosity bilayers (DOPC) are functionalized with HAVDI, this functionalization does not affect FA formation (Fig. 3c−e, Supplementary Fig 8a).

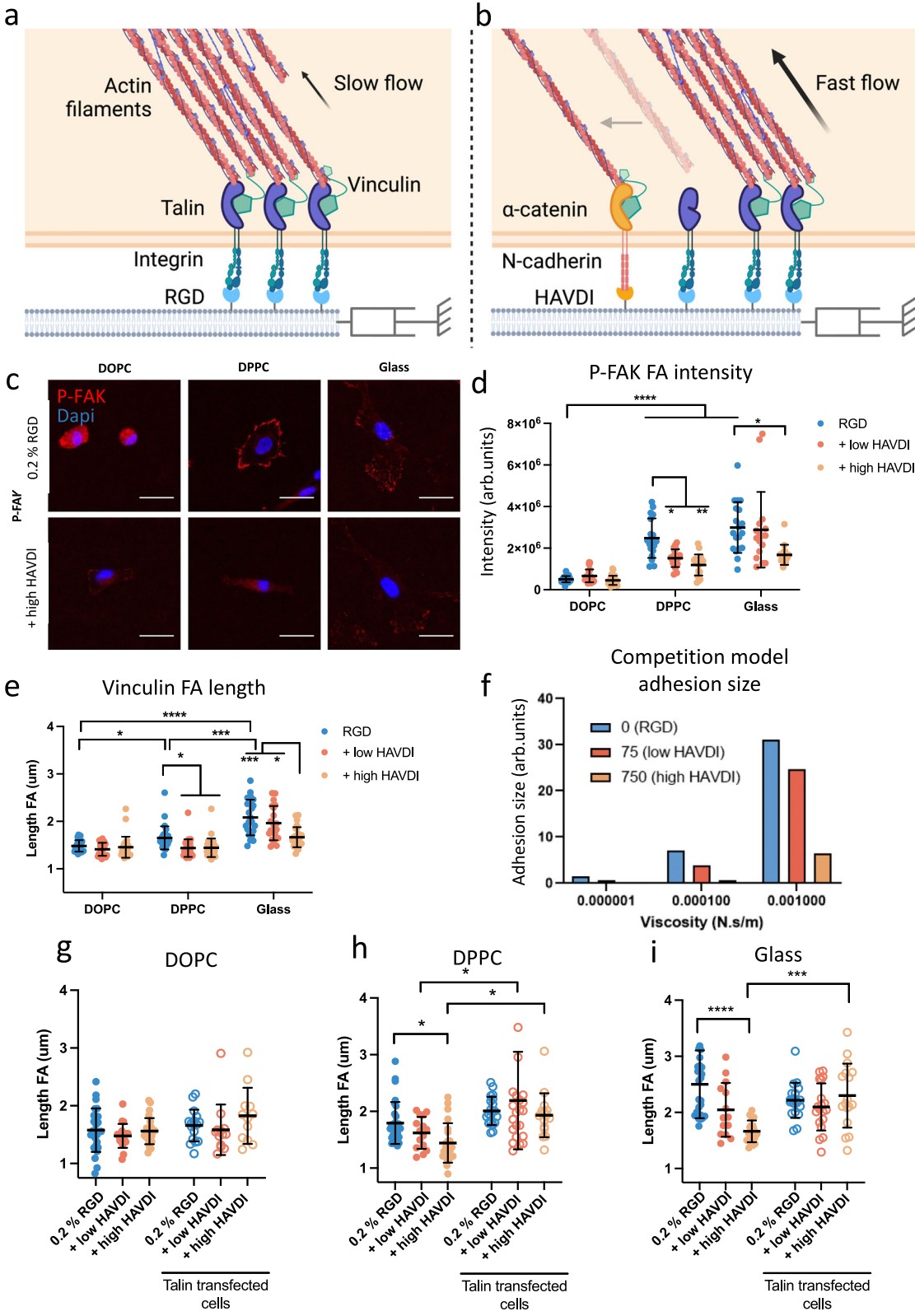

We also carried out experiments to test the hypothesis that our results can be explained by competition between RGD and HAVDI for actin binding. First, if the effect of this competition occurs at the level of actin binding, then it should be regulated not only by the concentrations of RGD and HAVDI but also by the concentrations of talin and α-catenin. To test this, we transfected Y201 human mesenchymal stem cells with the EGFP-talin1 plasmid to increase talin expression.

Increased talin expression should promote the formation of integrin-talin-actin clutches, rather than that of cadherin-catenin-actin, thereby reducing the effect of HAVDI. We used the hTERT Y201 immortalised MSC line for these experiments, instead of primary hMSCs, due to their higher resistance to mechanical stress and ease of transfection. hTERT Y201 MSCs behave in a similar way to primary hMSCs, being mechanosensitive and with a similar differentiation potential

**Fig. 3 | Adhesive crosstalk affects cell adhesion and the molecular clutch in a viscosity-dependent manner.** Schematic representation of the competition for actin filaments by integrin (**a**) and N-cadherin (**b**) adhesion clutches. Images made with BioRender. **c** Representative images of focal adhesions of hMSCs stained for P-FAK seeded on SLBs and glass with 0.2% RGD or 0.2% RGD plus high HAVDI (scale bar = 50 μm). **d** P-FAK intensity of hMSCs seeded on bilayers for 24 h with RGD and RGD plus low or high HAVDI (from left to right $n$ = 21, 22, 27, 18, 19, 20, 19, 17, 25). **e** FA size measured by vinculin staining of hMSCs seeded on bilayers for 24 h with RGD and RGD plus low or high HAVDI (from left to right $n$ = 23, 23, 29, 29, 30, 30, 23, 20, 26). Representative images can be seen in Supplementary Fig. 8b. **f** Adhesion size obtained by applying the molecular clutch competition model. Predictions are

shown for different amounts of HAVDI ligands (0-75-750) at three values of viscosity ($10^{-6}$ Ns/m for DOPC; $10^{-4}$ Ns/m for DPPC, and the highest modelled value of $10^{-3}$ Ns/m); full model predictions are shown in Supplementary Fig. 15. **g–i** FA length of Y201 overexpressing Talin seeded on DOPC, DPPC and glass, respectively, functionalized with RGD and low or high HAVDI (from left to right $n$ graph (**g**) = 26, 19, 25, 17, 13, 12; graph (**h**) $n$ = 29, 15, 23, 20, 23, 16; graph (**i**) $n$ = 25, 14, 19, 19, 19, 17). All statistical differences can be found in Supplementary Tables 5–7. Statistical significance was determined using D'Agostino Pearson normality test, followed by followed by a mixed-effects analysis. All data are presented as mean values ± SD. In all graphs $n$ represent cells. *$P \le 0.05$, **$P \le 0.01$, ***$P \le 0.001$, ****$P \le 0.0001$.

(Supplementary Fig. 9)[53]. We observed that increased talin expression led to more vinculin being recruited to the site of FAs, and to higher adhesion stabilization and increased clustering of integrins[54], favouring the binding of actin to RGD-based adhesions instead of N-cadherin ones (Fig. 3g–i). In these conditions, as expected and in contrast to what we observed in non-transfected cells on DPPC and Glass, the length of FAs in transfected Y201 cells was not affected by HAVDI (Fig. 3h, i). Higher levels of N-cadherin expression in the presence of HAVDI were instead still maintained (Supplementary Fig. 10 a).

We next regulated the competition for actin binding by stabilizing actin filaments using jasplakinolide, which impairs actin depolymerization[55]. In these conditions, increased actin availability should reduce competition between RGD and HAVDI for actin binding, thereby increasing integrin-mediated mechanotransduction and decreasing the effects of HAVDI ligands. Consistent with this hypothesis, jasplakinolide increased FA length in Y201 cells on DOPC substrates (Supplementary Fig. 10b, c). Importantly, when jasplakinolide was added to Y201 cells, the decrease in FA length that we observed in the presence of low or high HAVDI was abrogated on DPPC.

To further explore the mechanisms regulating the integrin-cadherin crosstalk, cadherins were also engaged at locations distinct from integrin-based interactions by using an anti-N-cadherin antibody to activate dorsal N-cadherins after cells had adhered (ventrally) on RGD-functionalised substrates. Crucially, we observed that dorsal cadherin engagement demonstrated through changes in β-catenin signalling (Supplementary Fig. 11a), did not lead to a decrease in FA formation on DPPC or on glass, as HAVDI functionalisation instead does (Supplementary Fig. 11b, c). These observations support the critical role of local competition for actin binding in the integrin-cadherin crosstalk.

To further investigate the role of adhesive crosstalk in viscosity mechanosensing, we used single cell force spectroscopy (SCFS) to study early adhesion forces in Y201 cells[56]. We observed an increase in the force of adhesion in Y201 cells at higher viscosities on RGD-functionalised surfaces, with the adhesion strength being highest on glass (up to 6 nN) (Fig. 4a and Supplementary Fig. 12a, b; Supplementary Fig. 12c, d for primary hMSCs). When Y201 cells were in contact with surfaces that also contained low or high HAVDI, we observed a decrease in the force of adhesion (Fig. 4a, b; Supplementary Fig. 12e, f for primary hMSCs). These results show that the competition between HAVDI and RGD also occurs during the early phases of adhesion formation. This eventually leads to the formation of smaller FAs on HAVDI-containing surfaces (Fig. 3e, Supplementary Fig. 9b), on which cells apply lower forces to the substrate through these complexes (Fig. 3d)[57].

We next measured the velocity of the rearward actin flow[58] in Y201 cells to investigate further the close relationship between FA formation, actin flow and the engagement of the molecular clutch. Our results show that actin flows were lower in more viscous surfaces in the presence of RGD (Fig. 4c, e). We propose that this decrease in velocity was caused by a higher engagement of the clutch by cells on less mobile RGD ligands, leading to a change in the conformation of talin, which exposes the vinculin adhesion site. Vinculin binds to talin, which

reinforces the FAs and promotes the formation and growth of actin stress fibres. At the same time, the myosin machinery pulls these fibres towards the cell nucleus. However, due to the strong adhesion of integrins to RGD ligands, the actin rearward flow slows down, as observed. This is also illustrated by the computational clutch model implemented here, which predicts a decrease in actin flow at increasing viscosity due to the engagement of the clutch (Fig. 4d)[12]. When we added low or high HAVDI to the RGD-functionalised surfaces, we observed faster actin flows compared to RGD-only surfaces (Fig. 4c, e). The modified computational clutch model predicts this increase in actin flow velocity in the presence of HAVDI at higher viscosity (Fig. 4d). Indeed, N-cadherin engagement disrupts the clutch due to HAVDI-based adhesions competing for actin binding with the integrin-based adhesions, which are reinforced only at high viscosity.

Actin flow measurements further supported our actin binding competition model for the integrin-cadherin crosstalk. Indeed, when cadherins were engaged dorsally via an anti-N-cadherin antibody instead of ventrally via HAVDI ligation, actin flow was not increased to the same levels (Supplementary Fig. 13). While a modest increase in actin flow was observed on DPPC, this was smaller than the one observed when ventral HAVDI ligation occurred, and, crucially, was not accompanied by a significant change in focal adhesion formation, as mentioned before (see Supplementary Fig. 11b). These observations confirm the local competition for actin fibres between integrins and cadherins as the key mechanism to regulate cross-talk following ventral HAVDI engagement. The observed slight variation in actin flow in response to administration of anti-N-cadherin antibody may have been elicited by some of the antibodies reaching the ventral side of the cells. Furthermore, overexpression of vinculin (a molecule involved in both integrin-talin and N-cadherin-α-catenin clutches) did not alter the HAVDI ligation-related increase in actin flow on DPPC and on glass (Fig. 4f, g, Supplementary Fig. 14). This supports that the key factor driving the competition between integrin and N-cadherin-clutches is actin rather than vinculin.

## Discussion

In vivo, hMSCs are in contact with both the ECM, via integrins, and with other cells, via N-cadherins. These adhesion receptors share several signalling pathway components, including cytoskeletal proteins[15,59,60]. As a result, mechanotransduction is affected by the 'adhesive crosstalk' that occurs between integrin-based and cadherin-based adhesions. In this work, we investigate the mechanosensing role of this integrin-cadherin crosstalk using a material platform based on bilayers of varying of viscosity. From our results, we propose that hMSC sense viscosity via RGD-based adhesions using a molecular clutch mechanism, in which the engagement of this clutch and the reinforcement of adhesion only occurs above a certain threshold of viscosity (Figs. 3f, 4d). When the adhesive crosstalk was elicited via HAVDI ligation, the most evident morphological effect we observed was a decrease in cell area compared to RGD-only surfaces (Fig. 1f). We also observed other effects on mechanosensing associated with N-cadherin ligation, such as a decrease in YAP nuclear translocation (Fig. 2b). Our results also show that viscosity sensing and adhesive crosstalk influence early

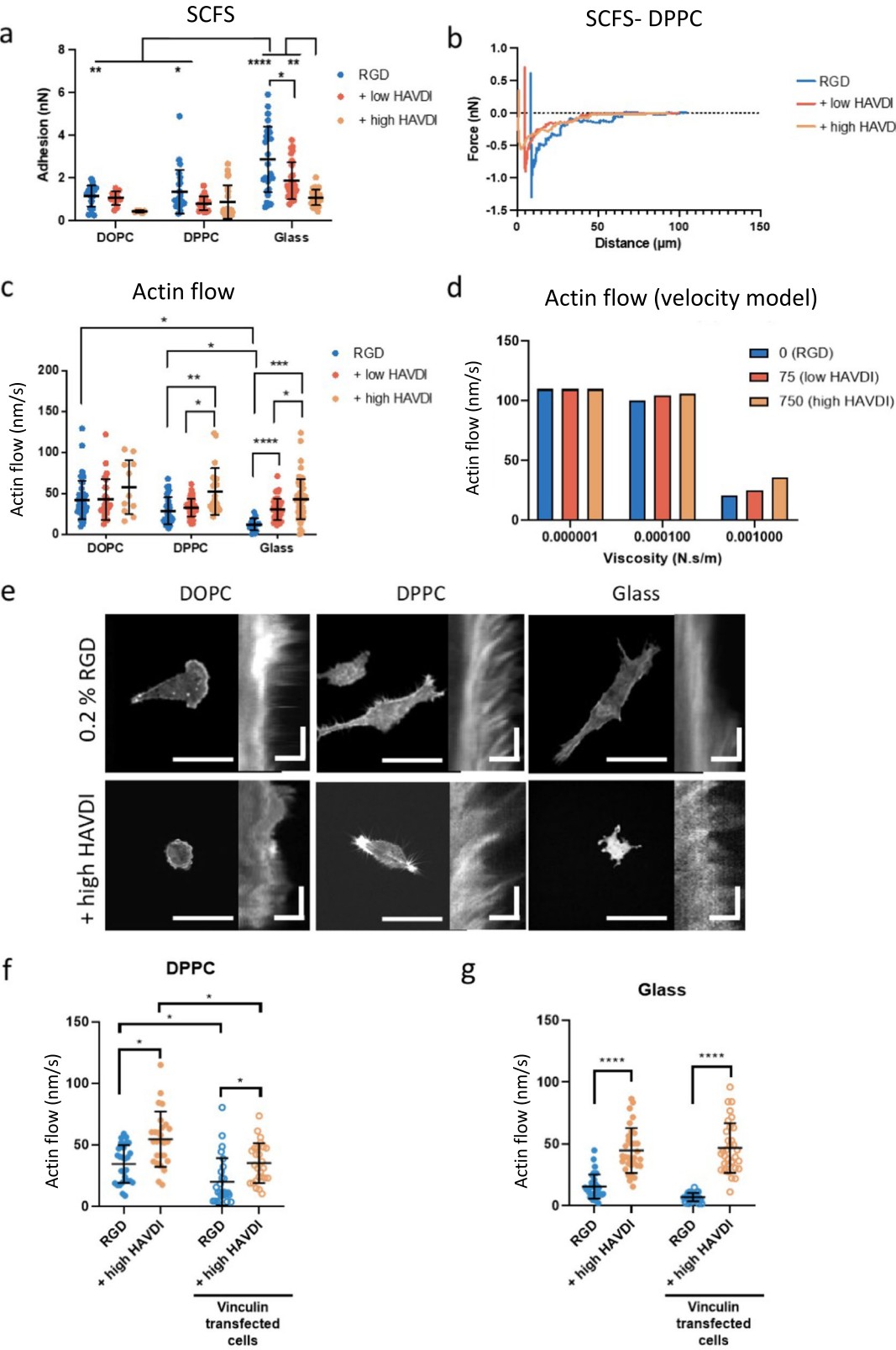

adhesive events, as measured by SCFS, after a few seconds of adhesion (Fig. 4a,b), as well as late events, such as hMSC differentiation (Fig. 2g–i).

On the basis that α-catenin-vinculin complexes in cadherin-based adhesions are analogous to talin-vinculin complexes in integrin-based adhesions, we interpret our results using a modified molecular clutch model, in which these complexes compete to bind to actin filaments.

This modified model predicts a viscosity-dependent decrease in FA size (Fig. 3f) and an increase in actin flow (Fig. 4d) when cells interact with different amounts of HAVDI in the presence of RGD, due to an increase in weaker HAVDI-mediated bonds. Our experimental results indeed show a reduction in FA length and an increase in actin flow on DPPC upon HAVDI ligation; these changes were not observed on low viscosity surfaces (DOPC) where the clutch was not engaged

**Fig. 4 | HAVDI ligation affects the force of adhesion and actin flow. a** Single-cell force spectroscopy of cells (Y201) seeded on bilayers or glass functionalized with RGD or RGD with low and high HAVDI (from left to right $n = 21, 12, 6, 24, 21, 28, 29, 28, 30$) $n$ represent force curves measured in 6 cells). Statistical significance was determined using D'Agostino Pearson normality test, followed by followed by a mixed-effects analysis. **b** Representative SCFS curves of cells seeded on DPPC bilayers functionalized with RGD or RGD with low and high HAVDI. **c** Actin flow of cells (Y201) seeded on supported lipid bilayers and glass functionalized with RGD or RGD with low and high HAVDI (from left to right $n = 47, 28, 11, 40, 59, 22, 15, 52, 58$) $n$ represents kymographs measured in 6 cells. Statistical significance was determined using D'Agostino Pearson normality test, followed by followed by a mixed-effects analysis. **d** Theoretical actin flow velocity obtained by applying the competition in the molecular clutch model. Predictions are shown for different

amounts of HAVDI ligands (0-75-750) at three values of viscosity ($10^{-6}$ Ns/m for DOPC; $10^{-4}$ Ns/m for DPPC, and the highest modelled value of $10^{-3}$ Ns/m); full model predictions are shown in Supplementary Fig. 15. **e** Representative images of cells transfected with LifeAct and their corresponding kymographs. **f, g** Actin flow of cells (Y201) seeded on DPPC bilayers and glass functionalized with RGD or RGD with high HAVDI (from left to right (graph **f**) $n = 27, 26, 27, 26$, (graph **g**) $n = 33, 32, 32, 33$). In graphs (**f** and **g**), $n$ represent kymographs measured in 5–6 cells. Scale bars: cell images = 50 μm; kymographs = 10 μm horizontal and 30 s vertical. Statistical significance was determined using D'Agostino Pearson normality test, followed by followed by (graph **f**) Kruskal-Wallis multiple-comparisons test or a (graph **g**) Two-Way ANOVA. All data are presented as mean values ± SD.*$P \leq 0.05$, **$P \leq 0.01$, ***$P \leq 0.001$, ****$P \leq 0.0001$.

(Figs. 3d, e, 4c). Results derived from the overexpression of talin and from the dorsal activation of N-cadherins also support our competition hypothesis, as talin overexpression abrogated the effect of N-cadherin on high viscosity surfaces alone (Fig. 3g–i) and N-cadherin engagement at locations distinct from integrins did not elicit the same response as ventral HAVDI ligation did (Supplementary Figs. 11 and 13).

The hMSC niche in vivo is a complex 3D environment in which integrins and cadherins engage with ECM proteins and cells of varying viscoelastic properties, an environment that cannot be fully recapitulated in simplified material platforms. Nevertheless, the platforms developed in this study provide us with essential tools with which to isolate and understand the effects of specific physical and biological cues. By using biointerfaces with controlled viscosity and ligand presentation, our work elucidates and models the molecular mechanisms that govern the adhesive crosstalk in response to viscosity. This approach not only furthers our understanding of how cells interact and respond to their environment but also informs the design of new synthetic substrates, which have applications that span from tissue engineering to the manufacturing of tissue models.

## Methods
### Preparation of lipid bilayers
Lipid bilayers were synthesised as in ref. 12. Briefly, 2 mg of either DOPC (1,2-dioleoyl-sn-glycero-3-phosphocholine, Avanti Polar Lipids, AL, USA) and DPPC (1,2-dipalmitoyl-sn-glycero-3-phosphocholine, Avanti Polar Lipids, AL, USA) were added to a glass vial together with B-cap-PE (1,2-dipalmitoys-sn-glycero-2-phosphoethanolamine-N-(cap biotinyl), (Avanti Polar Lipids, AL, USA) in different quantities to achieve 0.02, 0.2, 0.22, 2 or 2.2% of functionalization. Here, the amount of added biotinylated lipids was changed depending on the required functionalisation density to be obtained on each surface. DOPC and DPPC have a transition temperature of −17 °C and 41 °C respectively. When these lipids are under cell culture conditions (i.e. 37 °C) they are in a fluid phase in the case of DOPC and gel phase for DPPC. Both lipids were provided in chloroform solutions of 25 mg/ml. B-cap-PE was dissolved as suggested by the company. For regular usage, a more diluted solution was used at a concentration of 0.5 mg/ml, in 1.9 ml of chloroform and lipid solution, and 100 μl of methanol. Lipid mixtures were dried under $N_2$ gas and excess chloroform was removed by drying under vacuum for ≥1 h.

Lipids were rehydrated in rehydration buffer (150 mM NaCl and 10 mM Tris, pH 7.4) to a concentration of 2 mg/mL, allowed to swell above their transition temperature (DOPC $T_m = −19$ °C, DPPC $T_m = 41$ °C) and extruded through a 50 nm polycarbonate membrane (Whatman® Nucleopore Track-etched membrane, Avanti Polar Lipids, AL, USA) in the case of DOPC and through a 100 nm and 50 nm (above their Tm) for DPPC and stored for a maximum of 1 week.

### Preparation of glass surfaces
Borosilicate glass coverslips of thickness number 1 with 32 mm of diameter were cleaned using a mix of ultrapure water, ammonium

hydroxide and hydrogen peroxide (5:1:1) and heated up to 65 °C for 20 min. After that, they were washed with ultrapure water, dried with $N_2$ and stored in a sealed container until further use. Poly-dimethylsiloxane (PDMS) was used to create a confined space on the glass substrate and make the process of washing and cell seeding easier. The PDMS elastomer (Sylgard 184, Farnell, UK) was mixed in a 9:1 ratio with the crosslinker. The mixture was poured into a flat plastic dish and degassed under vacuum for 30 min or until there were no bubbles and cured in the oven for 2 h at 65 °C. Wells were created with a 9 mm stamp and bonded to cleaned glass using a handheld plasma device on the surfaces for 20 s. The surfaces were sterilized by exposure to UV light for 20 min prior use.

### Production and functionalization of SLBs and glass control
Lipid vesicles were diluted in F (fusion) buffer (300 mM NaCl, 10 mM Tris and 10 mM $MgCl_2$) to a final concentration of 0.1 and 0.2 mg/ml for DOPC and DPPC respectively and kept above $T_m$. Both solutions were sterilized by filtering through a 200 nm membrane. Glass surfaces were activated by oxygen plasma for 10 min in fixed cell experiments (Diener Electronics, 150 W) and 20 min for live cell experiments (Harrick Scientific, high power) and 150 μL of the vesicle solution was added to each well. Samples were incubated for 20 min above $T_m$ and washed extensively with F buffer and DPBS.

Functionalization of all samples was done using the biotin-avidin interaction. Firstly, neutravidin at 0.1 mg/ml (Fisher Scientific, USA) was added to the surfaces and incubated for 30 min. After washing with PBS, biotinylated ligands (RGD, alone or mixed with HAVDI or scrambled HAVDI in appropriate ratios) were added at 2 μg/mL, incubated for 30 min and washed with PBS and cell culture media (Genscript, Hong Kong). In this way, while the amount of biotinylated lipid in the bilayers controlled the amount of functionalization, the ratio between ligands controlled their final concentration (e.g. for a functionalization with 0.2% RGD and 0.02% (low) HAVDI, the biotinylated lipid is initially added at 0.22%, and ligands are finally added at a ratio RGD:HAVDI 10:1; for a functionalization with 0.2% RGD and 2% (high) HAVDI, the biotinylated lipid is initially added at 2.2%, and ligands are finally added at a ratio RGD:HAVDI 1:10).

In the case of glass, a biotin-PEG-silane (Silane-PEG-biotin, Nanocs, USA) was dissolved at a concentration of 10 mg/ml in 95 % ethanol 5 % water. The desired amount of silane was deposited onto a glass coverslip into a vacuum chamber with the glass-PDMS surfaces. The vacuum pump was turned on during 15 min and the samples were left under vacuum for 2 h. The surfaces were then sterilised under UV light for 30 min. NA functionalization was achieved following the same strategy previously mentioned but increasing 5 times the concentration. The surfaces were washed with PBS. 50 μL of a mixture of the biotin-PEG and Biotinylated peptides were then added (at a concentration of 10 μg/ml) and incubated for 30 min. Surfaces were washed thoroughly.

## Cell culture and Staining

hMSCs were seeded onto SLBs previously washed with DMEM (1x) Dulbecco's Modified Eagle Medium (+4.5 g/l D-Glucose, + L-Glutamine), supplemented with 100 µM sodium pyruvate, 1 % NEAA (non-essential amino acids), 1% of penicillin and streptomycin (P/S, Gibco) 1% Fungizone (Gibco) at a density of 10000 cells/cm² for fixed cell experiments and 20,000 cells/cm² for in-cell westerns assays. In the case of cell adhesion measurements, the pellet of cells is resuspended in CO₂ independent media (Gibco) and left in suspension until the measurement is done. hTERT Y201 was developed at the University of York and cultured with DMEM high glucose, sodium pyruvate and L-glutamine (Thermofisher) supplemented with 1% P/S. For immunostaining cells were cultured for 24 h. For longer experiments, media was supplemented with 10% FBS after 24 h. N-cadherin antibody (Sigma) was used for dorsal stimulation at a concentration of 100 µg/ml, introduced in cell media overnight, prior to experiment fixing. Cells were washed with DPBS and fixed with 4% paraformaldehyde and permeabilised with 0.1% Triton X-100. For immunostaining cells were blocked with 1% BSA for 30 min. Primary antibodies used were anti-vinculin (1:200, mouse monoclonal; Sigma-Aldrich), anti-P-FAK (1:200, mouse monoclonal; MilliPore), anti-YAP (1:100, mouse monoclonal; Santa Cruz Biotechnology), anti-N-cadherin (1:100, mouse monoclonal; BD biosciences), anti integrin $\alpha_v$ (1:40, mouse monoclonal, Abcam), anti-MIIA (1:400, rabbit monoclonal; Abcam). For In Cell Western cells were permeabilized with the permeabilization buffer (10.3 g of sucrose, 0.292 g of NaCl, 0.006 g of MgCl₂ and 0.476 h of HEPES, adjusted to pH 7.2 prior to adding 0.5 ml of Triton X) and blocked with 1 % milk powder for 1.5 h. Primary antibodies used were anti-RUNX2 (1:100, mouse monoclonal; Abgen), anti-P-RUNX2 (1:100, rabbit monoclonal; Arigo biolaboratories), anti-SOX9 (1:50, mouse monoclonal; Santa Cruz biotechnology), anti-PPARγ (1:100, mouse monoclonal; Santa Cruz biotechnology), anti-beta-catenin (1:200, mouse monoclonal; Abcam). Antibodies were incubated overnight and washed extensively with PBS. Samples were then blocked with respective blocking buffers and incubated with secondary antibodies for 1 h and washed 5 times for 5 min each. Secondary antibodies used for immunostaining were Cy3 rabbit anti-mouse (1:200, Jackson Immunoresearch), Alexa Fluor 568 (1:200, donkey anti-rabbit IgG, Invitrogen), Alexa fluor 488 phalloidin (1:100, Thermofisher). For ICW, IRDYE 680RD (1:500, Goat anti-mouse; 1:500), IRDYE 800RG (1:500, goat anti-rabbit, Li-Cor) and CellTag 700RD (1:800; Li-Cor) secondary antibodies were used. DAPI-containing mounting media was used to stain the nuclei.

Immunostaining images were taken with a Zeiss Observer.Z1 or Zeiss confocal 900 using Micro-Manager software or black ZEN respectively. Image processing and analysis were performed using Fiji imaging software. ICW samples were dried overnight and imaged with Odyssey Scanner at 700 and 800 nm.

## Focal adhesion quantification

FA size quantification was performed using the Fiji software. Firstly, the background was subtracted with a rolling ball radius of 50, then the contrast was enhanced through the CLAHE plugin. To further minimize the background, an exponential fit was run. Finally, the brightness and the contrast of the image were adjusted, and a threshold was applied. The particles were analysed by using the analyse particles function in Fiji.

In the case of FA intensity, a mask of the cell was made with the actin staining and the intensity of the cell was then calculated and normalized by the area.

## YAP quantification

In order to be able to analyse YAP, images of the actin cytoskeleton, nuclei and YAP were taken. Two stacks of images were done: one for the nuclei and other for YAP images. The nuclei were selected by thresholding, inverted, and normalized so that every nucleus had a grey value of 1 and the background 0. The background was subtracted from YAP images and multiplied by the thresholded, normalized corresponding nuclear image. The intensity of YAP in the nuclear region and in the whole cell was measured. The ratio was then calculated following Eq. (1).

$$YAP\frac{nuc}{cyt} = \frac{Intensity\ nucleus}{Intensity\ cell - Intensity\ nucleus} \tag{1}$$

Both the intensity of the nucleus and the intensity of the cytoplasm were normalized by the area of the correspondent region.

## Determining actin flow and Y201 transfection

Y201 cells were transfected using the Neon transfection system (ThermoFisher Scientific). Plasmids used were LifeAct-GFP (Ibidi), EGFP-talin 1 (gift from D. Critchley at University of Leicester, UK), Vinculin-GFP[61]. Parameters used to achieve cell transfection were 1400 V, 20 ms, 2 pulses, with 5 µg of DNA in both cases. Transfected cells were cultured for 24 h and used. To perform actin flow measurements on Vinculin overexpressing cells, cells were incubated with the SPY555-FastAct actin probe (Spirochrome) 2 h prior to imaging.

Glass bottom petri dishes were used for life imaging. Samples were cleaned by sonicating in ethanol for 30 min and activated with oxygen plasma for 20 min. Cells were seeded at 10,000 cells/cm² in DMEM for 3 h and imaged using a Nikon Eclipse Ti confocal spinning disk microscope with a 60x oil-immersion objective (N.A. 1.4) and Andor Q3 software. Images were taken for 4 min at 1 frame every 2 s at 488 or 555 nm. Actin flow was determined by kymographs in the Fiji software. In the case of talin transfection, cells were fixed and immunostained as previously described.

## Single-cell force spectroscopy

Bilayers were made as previously described by using a 200 µm mould of PDMS. One of the bilayers in the wells was made of only DOPC lipids in order to create a non-adherent surface to catch the cells. Cantilevers (0.03 N/m, ARROW-TL; nanoworld) were previously functionalized by immersing them in a 2 mg/ml concanavalin A solution overnight. Cantilevers were thoroughly washed with PBS and calibrated by recording a Force-Distance curve to determine sensitivity and spring constant. Y201 and hMSCs cells were captured by applying 10 nN set point, 70 µm pulling length, 5 µm/s extend speed and 15 s of contact with the cell. Cells were left to recover for 5 min and F-D curves were taken on the different surfaces for 30 s of contact time. Curves were taken using a Nanowizard 3 bioscience AFM (JPK, CA, USA) coupled with a Zeiss Axio Observer Z1 microscope and processed using JPK data processing software.

## Statistical analysis

Each experiment comprises three biological replicates. In all figures, values are given as the mean ± SD. Before statistical comparison, D'Agostino-Pearson omnibus normality test was performed to assess Gaussian distribution of the data. If the comparison was done between two populations t-test was performed. If more than 2 conditions were compared a one-way or two-way ANOVA test was performed. Statistical differences were defined by P values and confidence intervals were indicated with a *; ns > 0.05, * ≤ 0.05, ** ≤ 0.01, *** ≤ 0.001, **** ≤ 0.0001.

## Computational model

Details of the molecular clutch model are provided in the supplementary material (Supplementary Note 1).

## Reporting summary

Further information on research design is available in the Nature Portfolio Reporting Summary linked to this article.

## Data availability

All data supporting the findings in this study are available within the article and its Supplementary Information files; raw data can be obtained from the corresponding authors or can be accessed at the following repository [https://doi.org/10.5525/gla.researchdata.1769]. Source data are provided with this paper.

## Code availability

Code available from the corresponding authors (P.R.-C.) upon request.

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

## Acknowledgements

The work was supported by funding from the European Union's Horizon 2020 research and innovation programme (Grant agreement No. 874889—HEALIKICK), European Research Council AdG (101054728) and EPSRC through the Transformative Healthcare Technologies Programme Grant 'Mechanomeds' (EP/X033554/1). M.C. acknowledges funding from the Medical Research Council (MR/S005412/1) and the Royal Society (RGS/R1/231400). P.R.-C. Acknowledges funding from the Spanish Ministry of Science and Innovation (PID2022-142672NB-I00), the European Research Council (AdG 101097753), the Generalitat de Catalunya (2017-SGR-1602), and the prize 'ICREA Academia' for excellence in research. IBEC is member of CERCA Programme/Generalitat de Catalunya.

## Author contributions

E.B.-E. did the experimental work, analysed the data and wrote the original draft. M.A.G.O, F.C. and A.R.-N. contributed to the biological experiments. P.G. engineered and developed hTERT Y201 cells. M.J.D. contributed to the conceptual ideas, funding and edited the manuscript. P.R.-C., M.C. and M.S.-S. contributed to the conceptual ideas and funding, suggested experiments, led the development of the computational model and edited the draft to produce a final version. M.S.-S. takes responsibility on behalf of the authors with respect to the data, code and materials as described in the repository [https://doi.org/10.5525/gla.researchdata.1769].

## Competing interests

The authors declare no competing interests.
