## [Transparent Peer Review file · Nature Communications]

N-cadherin crosstalk with integrin weakens the molecular clutch in response to surface viscosity

Corresponding Author: Professor Manuel Salmeron-Sanchez

Version 1:

Reviewer comments:

Reviewer #1

(Remarks to the Author)

1. What are the noteworthy results?

This is a nice work showing how N-cadherin crosstalks with integrin to affect sensation of surface viscosity. Both the model and experiments are well designed and conducted, which implicate that cell-cell and cell-ECM adhesions compete with each other for actin cytoskeleton under the viscous environment.

2. Will the work be of significance to the field and related fields? How does it compare to the established literature? If the work is not original, please provide relevant references.

The significance of the work in relation to the field and related fields lies in its contribution to our understanding of how adherens junctions influence mechanosensation of viscosity. While previous research has established the crucial role of integrin-mediated focal adhesion in mechanosensation of viscoelasticity (Nature Materials, 2021, 20 (9): 1290-1299; Proceedings of the National Academy of Sciences, 2018, 115 (12): E2686-E2695), the present work expands upon this by investigating the impact of cadherins on integrin-mediated mechanosensation of viscosity. But, what are the differences in the antagonistic effects of the two receptors on the elastic and viscous environment, and what is the mechanical mechanism behind this difference?

To effectively situate the current work within the established literature, it is important for the authors to cite and compare the previous research on integrin-mediated focal adhesion and highlight the novel insights and observations provided by their study. By doing so, the authors can demonstrate the unique contributions of their work and provide a comprehensive overview of the existing knowledge in the field. This approach not only strengthens the credibility of the current study but also facilitates a clearer understanding of how it advances the current state of knowledge.

3. Does the work support the conclusions and claims, or is additional evidence needed?

Yes, the evidence is sufficient.

4. Are there any flaws in the data analysis, interpretation and conclusions? Do these prohibit publication or require revision? There are several flaws as list following:

(1) Quantitative Difference in Viscosity and Comparison between Experiments and Simulations:

It's crucial for the authors to provide a clear quantitative comparison of the viscosity of different supported lipid bilayers (SLBs) such as DOPC and DPPC, as well as glass. This comparison should include specific measurements and values to elucidate the differences in viscosity.

Additionally, direct comparisons between the results of experiments and simulations, particularly in Fig. 3e, f, Fig. 4c, and d, would enhance the robustness of the findings and strengthen the conclusions. Providing a side-by-side comparison of experimental and simulated data can help validate the simulation models and their relevance to the experimental observations.

(2) Physiological Relevance of Integrins and Cadherins in Sensing Viscosity:

The authors should explicitly discuss the potential in vivo scenarios where integrins and cadherins cooperate to sense the viscosity of the extracellular matrix (ECM). Exploring the physiological relevance of their findings will help establish the broader implications of the research in a biological context.

(3) Surface Density of Peptides, Functionalization Efficiency, and Theoretical Distance Among RGD Ligands:

It is imperative for the authors to provide detailed information about the surface density of peptides on functionalized SLBs and glass, along with the efficiency of functionalization processes. This information is critical for understanding the experimental setup and ensuring the reproducibility of the results.

Furthermore, the method and basis for obtaining the "theoretical distance among RGD ligands of 44 nm" should be clearly explained and supported by relevant references or experimental validation.

5. Is the methodology sound? Does the work meet the expected standards in your field?

Yes

6. Is there enough detail provided in the methods for the work to be reproduced?

Yes.

Reviewer #2

(Remarks to the Author)

This report explores the mechanism of cadherin - integrin crosstalk in response to mobility of their respective ligands. As reviewed properly in the manuscript, this cross-talk has been the subject of previous studies, including the specific ligands used in this study; what is new is a cytoskeleton-based competition mechanism to explain this phenomenon. The fundamental demonstration that engagement of cadherins reduces cell response to integrins, in a mobility-dependent manner, is overall strong. The subsequent examination of the underlying mechanisms is interesting, but incomplete. In summary, two issues - one technical and the other conceptual - limit my overall enthusiasm for this high-potential study. If addressed, this report has potential to advance the field by demonstrating a new mechanism of crosstalk between two signaling pathways.

1) Supported lipid bilayer stability. The cellular responses largely look at a 24 hour timepoint, which represents the integration of several complex molecular functions. Underlying much of this is the requirement that the substrate remain stable and intact over the experiment, which is not assured for supported lipid bilayers. Cells can displace or update lipid structures from such a surface. The extent of this type of interaction varies between cells and culture conditions. Some experiment showing that the lipid bilayer remains intact, with few holes through which proteins and cells can reach the underlying surface, is needed to address this issue. It should be certainly explored for the two lipid formulations with both ligands, but need not be an extensive experiment.

2) Mechanism of cross-talk. The proposed mechanism of competition between integrins and cadherins for actin fibers is interesting. However, and as reviewed in the manuscript, cadherin engagement modulates cytoskeletal dynamics through signaling pathways including Rac1 which can alter actin polymerization and receptor cluster formation, leading to complex and somewhat counterintuitive impacts on actin flow. The talin and Jaspilakinolide experiments are good steps, but don't address changes in these upstream pathways. Some measure or modulation of actin polymerization activity could help address this issue. Alternatively, engagement of cadherins at locations distinct from integrin interactions might rule out signaling-based effects as there would not be local competition for actin fibers.

Of minor note, there is a moderate level of typographical errors in the manuscript, including the Methods section. These do not dramatically impact the study, but should be addressed if this moves forward to publication.

Reviewer #3

(Remarks to the Author)

In this article, Barcelona-Estaje et al describe the effect of substrate viscosity on the crosstalk between N-cadherin and integrin adhesion sites. The article contains an impressive amount of data and some interesting results. However, in some cases, the conclusions drawn are not fully supported by the data and some experimental details, especially concerning the data analysis, are missing. As a general comment, I wonder if the 3 types of surfaces used to evaluate the effect of viscosity, these being DPPC, DOPC and glass, were the best possible choice to draw the conclusions that are present in the article. Importantly, the glass surface was functionalized with a concentration of neutravidin 5 times higher than the one used for lipid bilayers) and biotin-PEG was added, while in the lipid bilayers, only the biotinylated adhesion ligands were added. In most of the results shown, the difference between lipid bilayers and glass is larger than between DPPC and DOPC layers, and the glass control is often used to 'fix' the trend. My concern is that this surface is too different to be used to set the trend or make a conclusion regarding the increasing viscosity. It would have been more interesting to use a different lipid bilayer, for instance, DMPC, POPC or sphingomyelin layers. It would have also been interesting to see the results of having layers with HAVDI only.

In addition, I don't really see what the big conclusion of this study is. Is it that the viscosity influences the crosstalk between N-cadherin and integrin? How is this relevant for stem cells, or other cells? Are the values of viscosity used here related to biologically relevant values? Are the effects of the viscosity relevant in a biological setting, where many other cues are present?

More detailed comments on the manuscript can be found in the attached PDF.

Version 2:

Reviewer comments:

Reviewer #1

(Remarks to the Author)

The reviewer is satisfied with the revised manuscript and thus recommend publication of the manuscript in Nature Communications.

Reviewer #2

(Remarks to the Author)

The additional studies and narrative in this revised manuscript are much appreciated. However, they do not address the two concerns raised in my initial review. I cannot recommend this report for publication, based on the comments below.

1) Supported lipid bilayer stability.

The images of Supplementary Fig. 2i illustrate the stability problem. Cells are appearing in the fluorescence channel, indicating uptake of the BODIPY-functionalized lipids. The impact of removal of materials at some point will be production of holes, potentially below the limit of optical resolution, that will allow proteins from the media to attach to the surface. The representative image of Day 1 on DPPC in fact shows local depletion of lipids around the four adherent cells in the upper left quadrant of the image. A screenshot of this area with arrows indicating such regions is attached. The contrast has been increased to better highlight these issues.

If there is some other explanation for the cells appearing green in these images and the local depletion, it should be discussed in the narrative. Otherwise, reanalysis of key experiments where the analysis focuses on cells not exhibiting local disruption, is needed.

2) Mechanism of cross-talk.

The anti-N-cadherin experiments are interesting and much appreciated. However, Supplementary Fig. 13a shows that the anti-N-cadherin application does increase actin flow compared to RGD alone, particularly for DPPC. This could be due to the fact that the antibody could also be reaching the ventral side of the cells, as it is applied in solution. Regardless, addressing this change in actin flow is needed, whether through more extensive discussion on this mechanism on the overall impact of this study or additional experiments that more completely separate the signals. These experiments could include exposure of cells to beads coated with the anti-N-cadherin antibodies or using micropatterned surfaces with small regions of anti-N-cadherin interspersed into the lipid bilayer.

Reviewer #3

(Remarks to the Author)

I would like to extend my congratulations to the authors for their heroic effort in addressing all of my comments and remarks. All of my questions have been successfully addressed, and I believe this work is now fit for publication in its current state.

Version 3:

Reviewer comments:

Reviewer #2

(Remarks to the Author)

The revisions regarding lipid bilayer structure and impact on conclusions is appreciated. As noted, the stability issue is seen for many cell types, and remedying it would require a change in biological model or substrate design.

However, the term "reorganization" fails to capture that the cells are likely exposing part of the underlying substrate. I would recommend being more specific with "disruption" or "removal".

The revisions on cross-talk mechanism are also much appreciated, and have addressed my concern.

Response to reviewers

Reviewer #1:

1. What are the noteworthy results?

This is a nice work showing how N-cadherin crosstalks with integrin to affect sensation of surface viscosity. Both the model and experiments are well designed and conducted, which implicate that cell-cell and cell-ECM adhesions compete with each other for actin cytoskeleton under the viscous environment.

We thank the reviewer for their comments and feedback on our study, which have helped in improving our manuscript; please find our responses to their specific comments below.

2. Will the work be of significance to the field and related fields? How does it compare to the established literature? If the work is not original, please provide relevant references. The significance of the work in relation to the field and related fields lies in its contribution to our understanding of how adherens junctions influence mechanosensation of viscosity. While previous research has established the crucial role of integrin-mediated focal adhesion in mechanosensation of viscoelasticity (Nature Materials, 2021, 20 (9): 1290-1299; Proceedings of the National Academy of Sciences, 2018, 115 (12): E2686-E2695), the present work expands upon this by investigating the impact of cadherins on integrin-mediated mechanosensation of viscosity. But, what are the differences in the antagonistic effects of the two receptors on the elastic and viscous environment, and what is the mechanical mechanism behind this difference?

To effectively situate the current work within the established literature, it is important for the authors to cite and compare the previous research on integrin-mediated focal adhesion and highlight the novel insights and observations provided by their study. By doing so, the authors can demonstrate the unique contributions of their work and provide a comprehensive overview of the existing knowledge in the field. This approach not only strengthens the credibility of the current study but also facilitates a clearer understanding of how it advances the current state of knowledge.

The reviewer is correct that the novelty of the work lies in putting together integrins and cadherins in response to surface viscosity. Viscosity is an important component of viscoelasticity, a key property of the ECM (Nature 2020, 584: 535–546; <https://doi.org/10.1038/s41586-020-2612-2>). Yet, while significant amount of work has been done to understand cell response to elasticity, including the role of integrins and cadherins (Nature Materials 2016, 15(12): 1297–1306; <https://doi.org/10.1038/nmat4725>), the role of viscosity is not understood yet. We have expanded the introduction to include this.

3. Does the work support the conclusions and claims, or is additional evidence needed?

Yes, the evidence is sufficient.

We thank the reviewer for their support.

4. Are there any flaws in the data analysis, interpretation and conclusions? Do these prohibit publication or require revision?

There are several flaws as list following:

(1) Quantitative Difference in Viscosity and Comparison between Experiments and Simulations:

It's crucial for the authors to provide a clear quantitative comparison of the viscosity of different supported lipid bilayers (SLBs) such as DOPC and DPPC, as well as glass. This comparison should include specific measurements and values to elucidate the differences in viscosity. Additionally, direct comparisons between the results of experiments and simulations, particularly in Fig. 3e, f, Fig. 4c, and d, would enhance the robustness of the findings and strengthen the conclusions. Providing a side-by-side comparison of experimental and simulated data can help validate the simulation models and their relevance to the experimental observations.

We thank the reviewer for raising this point; we have previously quantified the viscosity of the different bilayers (Bennett et al., PNAS 2018; <https://doi.org/10.1073/pnas.1710653115>) to be 1×10^{-6} Pa·s·m for DOPC and 1×10^{-4} Pa·s·m for DPPC; the glass control, being not laterally mobile and essentially not deformable by the cell contractile machinery, is considered infinitely viscous in terms of viscosity. We have added this information in the revised version of the paper. We have modified the figures to facilitate a comparison between the trends predicted in the models and observed in the experimental data (Figures 3f and 4d). Specifically, we now show model data starting at a viscosity of DOPC, 1×10^{-6} Pa·s·m, and we represent the model predictions at three distinct viscosity values, representing DOPC, DPPC and glass (highest modelled viscosity); predictions for the full ranges of viscosity are now shown in Supplementary Figure 15. We note that an exact match between experiments and model is not expected, as the model is a highly simplified system. Instead, the model is meant to show that the fundamental interactions of a molecular clutch system, combined with competition between integrins and cadherins for actin, can predict the observed trends in actin flows and adhesions as a function of changes in RGD and HAVDI concentrations. Further, the predicted trends take place in a viscosity range that matches the order of measured DOPC and DPPC viscosities.

(2) Physiological Relevance of Integrins and Cadherins in Sensing Viscosity:

The authors should explicitly discuss the potential *in vivo* scenarios where integrins and cadherins cooperate to sense the viscosity of the extracellular matrix (ECM). Exploring the physiological relevance of their findings will help establish the broader implications of the research in a biological context.

The ECM is viscoelastic in nature and there is increasing interest in understanding cell response to it. Examples such as collective cell migration – that underpins development, cancer progression and regeneration – are explained through integrin and cadherin crosstalks in response to matrix stiffness. Yet, the ECM is viscoelastic and apart from a few seminal papers in the field, some excellent reviews have been published in the last few years (see e.g. Nature 2020, 584: 535–546; <https://doi.org/10.1038/s41586-020-2612-2>). Our work lies in the context of understanding the role of the viscous part of viscoelasticity and how this impacts integrins and cadherins. Integrins are key to interact with the ECM – of viscoelastic nature – whereas cadherins link cells and their role is modulated by the viscosity of the cell membrane (Adv

Healthc Mater. 2020;9(8):e1901259; <https://doi.org/10.1002/adhm.201901259>). This has been discussed in the revised version of the paper.

(3) Surface Density of Peptides, Functionalization Efficiency, and Theoretical Distance Among RGD Ligands:

It is imperative for the authors to provide detailed information about the surface density of peptides on functionalized SLBs and glass, along with the efficiency of functionalization processes. This information is critical for understanding the experimental setup and ensuring the reproducibility of the results.

Furthermore, the method and basis for obtaining the "theoretical distance among RGD ligands of 44 nm" should be clearly explained and supported by relevant references or experimental validation.

Peptide density is regulated via the amount of biotinylated lipid added in the lipid mixture; we have previously confirmed peptide densities via quantitative fluorescence microscopy (Bennett et al., PNAS 2018; <https://doi.org/10.1073/pnas.1710653115>). The theoretical distance between lipids is calculated as $1/\sqrt{n}$, where n is the particle density, as now explained in the manuscript.

5. Is the methodology sound? Does the work meet the expected standards in your field?
Yes.

6. Is there enough detail provided in the methods for the work to be reproduced?
Yes.

We thank the reviewer for their support.

Reviewer #2:

This report explores the mechanism of cadherin - integrin crosstalk in response to mobility of their respective ligands. As reviewed properly in the manuscript, this cross-talk has been the subject of previous studies, including the specific ligands used in this study; what is new is a cytoskeleton-based competition mechanism to explain this phenomenon. The fundamental demonstration that engagement of cadherins reduces cell response to integrins, in a mobility-dependent manner, is overall strong. The subsequent examination of the underlying mechanisms is interesting, but incomplete. In summary, two issues - one technical and the other conceptual - limit my overall enthusiasm for this high-potential study. If addressed, this report has potential to advance the field by demonstrating a new mechanism of crosstalk between two signaling pathways.

1) Supported lipid bilayer stability. The cellular responses largely look at a 24 hour timepoint, which represents the integration of several complex molecular functions. Underlying much of this is the requirement that the substrate remain stable and intact over the experiment, which is not assured for supported lipid bilayers. Cells can displace or update lipid structures from such a surface. The extent of this type of interaction varies between cells and culture conditions. Some experiment showing that the lipid bilayer remains intact, with few holes through which proteins and cells can reach the underlying surface, is needed to address this issue. It should be certainly explored for the two lipid formulations with both ligands, but need not be an extensive experiment.

We agree with the reviewer about the importance of demonstrating the bilayer's stability during culture. We have added a new Supplementary Figure 2i, which shows bilayer stability after 1 and 5 days in the presence of cells; this is done through observation of bilayers containing 0.1 mol% BODIPY-functionalized lipids.

2) Mechanism of cross-talk. The proposed mechanism of competition between integrins and cadherins for actin fibers is interesting. However, and as reviewed in the manuscript, cadherin engagement modulates cytoskeletal dynamics through signaling pathways including Rac1 which can alter actin polymerization and receptor cluster formation, leading to complex and somewhat counterintuitive impacts on actin flow. The talin and Jaspilakinolide experiments are good steps, but don't address changes in these upstream pathways. Some measure or modulation of actin polymerization activity could help address this issue. Alternatively, engagement of cadherins at locations distinct from integrin interactions might rule out signaling-based effects as there would not be local competition for actin fibers.

We thank the reviewer for raising this interesting point. As suggested, we have engaged cadherins at locations distinct from integrins using an anti-N-cadherin antibody to activate dorsal N-cadherins after cells have adhered on the bilayers. We observed that dorsal cadherin engagement (demonstrated through changes in β -catenin signalling) does not lead to a decrease in FA formation on DPPC or glass, as HAVDI-functionalised bilayers instead do. This is shown in new Supplementary Figure 11. Similarly, actin flow is not increased to the same levels as on HAVDI-functionalised bilayers (new Supplementary Figure 13). These observations support the talin overexpression and the jaspilakinolide experiments in pointing to the competition between integrins and cadherins for actin fibres as the main mechanism for cross-talk.

Of minor note, there is a moderate level of typographical errors in the manuscript, including the Methods section. These do not dramatically impact the study, but should be addressed if this moves forward to publication.

We apologise for these errors, which we have now corrected.

Reviewer #3:

In this article, Barcelona-Estaje et al describe the effect of substrate viscosity on the crosstalk between N-cadherin and integrin adhesion sites. The article contains an impressive amount of data and some interesting results. However, in some cases, the conclusions drawn are not fully supported by the data and some experimental details, especially concerning the data analysis, are missing. As a general comment, I wonder if the 3 types of surfaces used to evaluate the effect of viscosity, these being DPPC, DOPC and glass, were the best possible choice to draw the conclusions that are present in the article. Importantly, the glass surface was functionalized with a concentration of neutravidin 5 times higher than the one used for lipid bilayers and biotin-PEG was added, while in the lipid bilayers, only the biotinylated adhesion ligands were added. In most of the results shown, the difference between lipid bilayers and glass is larger than between DPPC and DOPC layers, and the glass control is often used to ‘fix’ the trend. My concern is that this surface is too different to be used to set the trend or make a conclusion regarding the increasing viscosity. It would have been more interesting to use a different lipid bilayer, for instance, DMPC, POPC or sphingomyelin layers. It would have also been interesting to see the results of having layers with HAVDI only.

We thank the reviewer for raising these issues as it has allowed us to clarify details regarding the material platform used in our study. In terms of viscosity and bilayers choice, we have used DOPC and DPPC because we have previously demonstrated that they provide a range of viscosity able to modulate the molecular clutch engagement (Bennett et al., PNAS 2018; <https://doi.org/10.1073/pnas.1710653115>). As in the previous study, we consider glass as an infinitely viscous substrate, with the ligands being not laterally mobile and essentially not deformable by the cell contractile machinery. On glass, a higher amount of neutravidin is used compared to the bilayers because the entire surface is coated with neutravidin, which is then bound to defined mixtures of biotin-PEG/biotin-ligand which determine the number of ligands available. On the bilayers the addition of biotin-PEG is instead not necessary because ligands’ amounts are controlled by the amount of biotinylated lipid in the bilayer. Using these strategies, the density of ligands is similarly controlled on both the glass surfaces and the bilayers. With regards to bilayers functionalised only with HAVDI, these are not able to support cell adhesion and for this reason they are not included.

In addition, I don’t really see what the big conclusion of this study is. Is it that the viscosity influences the crosstalk between N-cadherin and integrin? How is this relevant for stem cells, or other cells? Are the values of viscosity used here related to biologically relevant values? Are the effects of the viscosity relevant in a biological setting, where many other cues are present?

The reviewer is correct that we demonstrate that viscosity influences the crosstalk between N-cadherins and integrins. Further, we present a mechanism by which N-cadherin binding to ligands modulates integrin mediated-adhesion through competition for acting fibres that in turn reduce the force loading rate via the molecular clutch model. The study is relevant as viscoelasticity is a key property of the ECM and, e.g., stem cells differentiate to osteoblasts on soft hydrogels that maintain elasticity and increased viscosity (i.e. stress relaxation in Nature Materials 2016;15:326-34; <https://doi.org/10.1038/nmat4489>). Here, we decouple elasticity from viscosity to understand its role in building up integrin adhesion in an environment where cadherins are present. Of course, we fully agree with the reviewer that in a biological setting *in vivo* many other factors apart from viscosity would be present. However, isolating its

contribution, as done here, is useful to understand its potential role in any context where a viscous component is present.

More detailed comments on the manuscript can be found in the attached PDF.

1. I have some questions regarding the concentration of peptides used, and what they mean. In the methods section the authors wrote that biotinylated lipids were used in different quantities to achieve 0.02, 0.2, 0.22, 2 or 2.2 % of functionalization (correct?). Is this the limiting factor on the functionalization of the bilayers? For instance, for a bilayer with 0.2% RGD + 2% HAVDI, a lipid mixture containing 2.2% of biotinylated lipids is used and a solution of 1:10 of RGD:HAVDI is added to the solution? Can you please clarify this. Since the biotinylated lipids are always DPPC, will there be an effect on the viscosity between the layers with 0.02 and the 2.2% (for the DOPC samples)?

The reviewer is correct. The amount of biotinylated lipids controls the amount of functionalisation, and ligands are then added in the appropriate ratio. We apologise if this was not clear in our methodological section, and we have now addressed this. Varying the amount of biotinylated lipid within the explored ranges does affect bilayer mobility, as measured via fluorescence correlation spectroscopy and shown in the new Supplementary Figure 2j.

2. The authors state that they use 0.2% and 2% HAVDI concentration as the ‘low’ and ‘high’ HADVI. In supporting figure 4, 10% HAVDI was used. Can the authors provide some rationale to why the 0.2 and 2% HAVDI conditions were chosen? And how was the layer with 10% HAVDI prepared?

0.02% and 2% were chosen after an initial screening of concentrations ranging from 0.02% to 10%. The layer with 10% HAVDI was obtained by adding up to 12% biotinylated lipids in the lipid mixture (depending on the required RGD concentration).

3. For the quantification of the peptides in supplementary figure 3 I have a couple of comments. First, it would be useful to have a control with the intensities of the bilayers containing only streptavidin. Then we could also confirm the presence of RGD and HADVI at the lower concentration. Secondly, the image presented on panel seems brighter, while the quantification shows that there is no difference. Is this because of the LUT of the image or was this not a very representative image? Also, the amount of HAVDI is 10x higher but the fluorescence intensity detected is only 2-2.5x higher – any explanations? And, finally, since there are a lot of experiments presented in the paper with varying concentrations of RGD, I would also quantify the amount of RGD only in the layers, to be sure that there is an increase.

We thank the reviewer for raising these comments. The images with neutravidin only are not shown because there was no signal present. In terms of concentrations, the images for lower HAVDI concentration are shown in panels b) and f), and their intensity is similar. We have previously confirmed peptide densities using this lipid bilayer platform via quantitative fluorescence microscopy of fluorescent neutravidin (Bennett et al., PNAS 2018; <https://doi.org/10.1073/pnas.1710653115>). Here, we did not adopt a quantitative fluorescence microscopy technique and instead focussed on comparing single ligands with mixtures to ensure adequate and consistent functionalisation when ligand mixtures were used.

4. What do the author mean by ‘a higher intensity of N-cadherin expression’ (end page 5). The supplementary figure 3 shows the quantification of the antibody in panel a, but the raw images are not shown. How did the authors quantify this? Is this per cell, an average? Just a cross-section or the whole volume? Depending on how the images are analyzed, the difference can be related to the different cellular localization instead of an expression level.

We thank the reviewer for this comment. Images were analyzed by creating a mask of the cell and then using it to define the region of interest for the stained N-Cadherins. Once this was done, the intensity of the N-Cadherin staining was calculated per cell and normalized by the area. Images of the N-Cadherin staining have been added to the supplementary material (Supplementary Figure 3b).

5. “when lower concentrations (0.02 % mol) of RGD were tested (i.e. higher RGD spacing), changes in the size of hMSC area due to HAVDI were observed only for DPPC and not for DOPC” – it is not clear why are there differences between DOPC and DPPC in this case; please add some explanation.

At very low RGD concentrations and high intermolecular distances, cell attachment and cell spreading are minimal on DOPC. In this condition, the disruption introduced by the addition of HAVDI cannot have any further effect in reducing cell area. This is not the case on DPPC substrates, where cells can still spread despite the low amount of RGD ligands.

6. The data presented in figure 1e,f,g and in supplementary figure 4b is partially referent to the same experimental conditions but the trends shown are different. For instance, DPPC with 0.2%RGD, the cell area in supplementary figure 4 increases when going from 0.02 to 0.2% HAVDI, which is not in agreement with the trend discussed in the main text.

We thank the reviewer for this observation. In Figures 1e,f,g, we show low HAVDI, which corresponds to 0.02%, and high HAVDI, which corresponds to 2%; 0.2% HAVDI mentioned by the reviewer is not reported in this figure. In Supplementary Figure 4, we show varying HAVDI concentrations, from 0.02, to 0.2, 2 and 10%; the change from 0.02 to 0.2% HAVDI in Supplementary Figure 4b is not statistically significant.

7. Regarding the colocalization of MIIA with the focal adhesions, I think that the data does not support the conclusions in the manuscript. The authors stated “adding HAVDI decreases MIIA levels in the focal adhesions”. By looking at the images, it seems that the presence of HAVDI leads to a much higher expression of MIIA (brighter cells). It is difficult to see if there is MIIA present at the focal adhesions because the level in the cytosol is much higher. The authors show intensity profiles, but it is difficult to interpret these without knowing their location of the image. To retrieve some quantitative analysis, the authors could do some image correlation analysis, rather than showing one intensity profile for each condition.

We thank the reviewer for raising this point. Even though we agree that image correlation analysis would be ideal for this statement, it cannot be done with the current images presented

in this manuscript. In order to try to clarify this, we have added lines where the intensity profiles have been taken so that a clear correlation between figures and profiles can be established.

8. On figure 2, the data shown in panel b, and on panels c and d is different, while it is acquired for the experimental conditions (e.g. compare DPPC RGD only in panel b and panel d). Can the authors explain this? This is especially important since in panel d there is no decrease in the nuc/cyt ratio of YAP between RGD only and low HAVDI, while the authors claim a difference in panel b (with $p < 0.0001$)...

The decision not to include the comparisons between RGD and low and high HAVDI conditions in every panel was made to avoid overcrowding the graph and to ensure clarity of interpretation. As the data in Figure 2 panel b and panels c-e are the same experimental conditions for RGD and HAVDI, they have now been merged; merged data is shown in panel b. In panels c-e, only the conditions of interest for these panels (HAVDI or scrambled HAVDI) and the corresponding differences (HAVDI vs. scrambled), if present, are now indicated.

9. For the differentiation markers (figure 1g, h, i), to be able to draw conclusion on the effect of viscosity, it is necessary to calculate/show the statistical differences between the same condition in different substrates. For a first look, it seems that the data is too spread to be able to draw any conclusions (differences not statistically significant). This is valid for all the expression markers analysed.

We understand the reviewer's concern regarding the need to present statistical differences between the same conditions on different substrates. However, we would like to clarify that while we acknowledge the perceived spread in the data, we have indeed compared the data between conditions on different substrates. We would like to note that all statistical differences among conditions have been now included in Supporting information as Tables 2, 3 and 4.

10. "Interestingly, at increasing concentrations of HAVDI, SOX9 expression increased independently of viscosity." Except for the glass, where there seems to be a drop. Can you comment on this?

While HAVDI is known to promote chondrogenesis, there are reports of osteogenesis being induced in its presence (Zhu et al. 2016, *Biomaterials* 77, 44-52; <https://doi.org/10.1016/j.biomaterials.2015.10.072>). It can be argued that this is the case on glass, where an increase in pRUNX2 is observed. We have included this observation in the manuscript.

11. The results regarding the intensity and length of the focal adhesion are difficult to interpret due to the lack of information regarding the analysis. How is the intensity calculated? Is there any image segmentation performed? How?

We apologize for not including this in the text. In response to this query, we have incorporated detailed information regarding the calculation of intensity, as well as any image segmentation procedures, in the Methods section of our manuscript.

12. On p14 the authors claim "only for more viscous substrates, smaller FAs when adding HAVDI". The differences between RGD, low HADVI, high HAVDI should be quantified in all viscosity conditions (also graph 3e).

We apologise as we don't understand what the reviewer means here. Quantification of the length of FAs are shown in Figure 3e for all the conditions in the paper (i.e. RGD, low HAVDI and high HAVDI).

13. For Vinculin, only the quantification is shown (Figure 3e). Please add the fluorescence images as well. Add also the images, for FAK and vinculin, for low HAVDI (even if just in SI).

We thank the reviewer for this comment. Representative images for all the conditions have been added as Supplementary Figure 8b.

14. The authors claim that "by overexpressing talin, FAs increase." They should calculate statistical significances between each condition for non-transfected and transfected cells.

We have considered the reviewer's suggestion and statistical significances between each condition for non-transfected and transfected cells have been calculated and added to the relevant sections of the revised manuscript. We also note that all statistical differences for transfected and not transfected cells have been included in the supporting information as Tables 5, 6 and 7.

15. The authors claim that "as expected and in contrast to what we observed in control cells on DPPC, the length of FAs in transfected Y201 cells was not affected by HAVDI" and "as expected and in contrast to what we observed in control cells (= no transfection I assume) on DPPC AND GLASS (?), the length of FAs in transfected Y201 cells was not affected by HAVDI". The authors should show statistical differences between RGD-low-high in transfected cells in all conditions.

We thank the reviewer for this comment. Following their suggestion, we have now included the result of the statistical analyses comparing RGD-low-high conditions in transfected cells across all experimental conditions. We note that all statistical differences have been added in the supporting information as Tables 5, 6, and 7.

16. "In this model, talin-vinculin and α -catenin-vinculin would compete with each other to bind to actin, given the limited availability of actin filaments". Could this also be related to the availability of vinculin?

The reviewer has raised here a very important point that in essence suggests an alternative hypothesis to the one we included in our manuscript, i.e. whether the competition between integrins and N-cadherins could be for vinculin instead of actin filaments. To demonstrate this, we performed additional experiments measuring the actin flow on DPPC and glass with RGD and RGD+HAVDI using wild type cells and also cells that have been transfected to overexpress

vinculin. Results are shown in Figure 4f-g and Supplementary Figure 14 and demonstrate that, even in the transfected cells, the actin flow increases for cells on RGD+HAVDI, supporting that the competition is actually for actin filaments. We thank the reviewer for this insightful comment that made us perform additional experiments that validate our hypothesis and strengthen the manuscript.

17. “When low-viscosity bilayers (DOPC) are functionalized with HAVDI, this functionalization does not affect FA formation”. Can the authors provide some hypothesis as to why this is happening?

The overarching message of this manuscript is that when the molecular clutch is engaged, then the presence of HAVDI leads to a weakening in cell adhesion as reflected by a decrease in the size of focal adhesions and increase in the actin flow. However, the molecular clutch is not engaged on DOPC when the substrate is functionalised with RGD (as already demonstrated in Bennett et al., PNAS 2018; <https://doi.org/10.1073/pnas.1710653115>). Therefore, the effect of adding HAVDI on DOPC substrates does not alter focal adhesion formation, which is already minimal before HAVDI addition.

18. “hTERT Y201 MSCs behave in the same way as primary hMSCs, being mechanosensitive and with a similar differentiation potential (Supplementary Figure 9)”. I disagree with this statement. If the authors want to claim this, they should put both cell lines next to each other in the same graphs. Supplementary figure 9 is only about Y201 cells. For example SOX9 levels seem quite different in both cell lines: primary cells have SOX9 0.05-0.15 (fig 2h) while Y201 have 2-4 (fig S9h). Also, the trend between the RGD-low-high for DPPC and glass are different between the 2 cell lines. And the cyt/nuc ratio for YAP is very different, suggesting that the cells ARE responding differently.

We are sorry for the misrepresentation of the Y201 cells in this statement. We meant to say that hTERT Y201 MSCs behave in a *similar way* as primary hMSCs, maintaining mechanosensitivity and differentiation potential. We have made this clear in the revised version of the manuscript.

19. “We observed that increased talin expression led to more vinculin being recruited to the site of FAs”. So maybe vinculin is drawn away from N-cadherin adhesions, and this is the rate-limiting step (not actin)? This claim could be support by imaging N-cadherin with talin overexpression to check if there was less cadherin present.

This comment is related to number 16 before, where the reviewer explores the idea of whether the limiting factor is vinculin instead of actin. Further to the data already mentioned in comment 16 (i.e., actin flow experiments with cells transfected to overexpress vinculin), we have also performed additional experiments to show N-cadherin staining in cells transfected to overexpress talin. In agreement with our hypothesis, high levels of HAVDI still leads to higher expression of N-cadherin for the transfected cells. This is now shown in Supplementary Figure 10a.

20. Add more information on the quantification and statistical analysis. For instance, in many cases, the total number of observations is not mentioned, nor the number of biological replicates.

We thank the reviewer for this comment. We have added the number of observations in the figures. The number of biological replicates for all figures in the revised version of the manuscript has been added in the methods section.

21. % mol, mol %, % are used interchangeably, make more uniform throughout the Manuscript

We apologize about the change of nomenclature. It has been homogenised along the text in the revised version of the manuscript.

22. Please add also the lower error bars in the bar graphs (in both the manuscript and the SI)

Lower error bars in the graph have been added in the revised version of the manuscript (including Supporting Information).

23. The letters and text describing the viscosity values in figure 1a are too small to read. Can you also mention the source of these values or how were they obtained?

We apologise for this. The size of the text in figure 1a has been increased and their source has also been included in the revised version of the manuscript.

24. Figure 1c, there seems to be no data for the non-functionalized surface (no error bar visible)

On non-functionalized DOPC, there were no cells on any sample, hence the cell density value was 0. This has been made clear in the revised version of the manuscript.

25. For figure 1h, it would be better to have an overview image (similar to that shown in sup. Fig. 3b). If you opt for showing the individual cells, please center the images...

We appreciate the reviewer's observation and overview images have been included for figure 1h in the revised version of the manuscript.

26. Scale bars: either always in the figures or in the caption (eg. fig 1b versus 1h). Also, please use μm instead of um.

This has now been corrected.

27. Mention supplementary figure 5 in the main text.

Supplementary Figure 5 is now mentioned in the “N-cadherin ligation affects hMSC mechanosensing and differentiation” section.

28. Add the time scales for the kymographs in figure 4.

We appreciate this comment and the time scales have been added in the kymographs in Figure 4.

29. “at a density of 10000 cells/cm² for fixed cell experiments and 20000 cells/cm² for in-cell westerns assays.” Why the different densities? Does this have an impact on the results?

We used higher cell densities for In-Cell Western assays because of their low sensitivity, requiring higher numbers of cells to get a measurable signal.

30. Please review the methods section of typos (there are a LOT).

We apologise for the typos, which have been corrected throughout the manuscript.

Response to reviewers

Reviewer #1:

The reviewer is satisfied with the revised manuscript and thus recommend publication of the manuscript in Nature Communications.

We are pleased that the reviewer is satisfied and recommends publication of this manuscript.

Reviewer #2:

The additional studies and narrative in this revised manuscript are much appreciated. However, they do not address the two concerns raised in my initial review. I cannot recommend this report for publication, based on the comments below.

1) Supported lipid bilayer stability.

The images of Supplementary Fig. 2i illustrate the stability problem. Cells are appearing in the fluorescence channel, indicating uptake of the BODIPY-functionalized lipids. The impact of removal of materials at some point will be production of holes, potentially below the limit of optical resolution, that will allow proteins from the media to attach to the surface. The representative image of Day 1 on DPPC in fact shows local depletion of lipids around the four adherent cells in the upper left quadrant of the image. A screenshot of this area with arrows indicating such regions is attached. The contrast has been increased to better highlight these issues.

If there is some other explanation for the cells appearing green in these images and the local depletion, it should be discussed in the narrative. Otherwise, reanalysis of key experiments where the analysis focuses on cells not exhibiting local disruption, is needed.

The reviewer questions the stability of the substrates and has pointed out the local depletion of lipids around cells. This is a point that we did not note in the previous version of the manuscript and so we need to thank the reviewer again for bringing it up. It is known that cells remodel proteins at the cell-material interface and that when proteins are loosely attached to the substrate they can be 'removed' from it and eventually internalised, e.g. fibronectin on glass (see the pioneering work of Grinnell and Geiger: *Cell*, 25, 121-132 (1981) [https://doi.org/10.1016/0092-8674\(81\)90236-1](https://doi.org/10.1016/0092-8674(81)90236-1) and *J Cell Biol* 103, 2697 (1986) <https://doi.org/10.1083/jcb.103.6.2697>; and the work of others such as Altankov *J Biomed Mater Res* 30, 385 (1996) [https://doi.org/10.1002/\(SICI\)1097-4636\(199603\)30:3<385::AID-JBM13>3.0.CO;2-J](https://doi.org/10.1002/(SICI)1097-4636(199603)30:3<385::AID-JBM13>3.0.CO;2-J)). This mechanical remodelling at the interface determines the compatibility of substrates. Insomuch that when mechanical remodelling does not happen because, e.g., proteins are strongly attached to the underlying material then cells increase protease secretion and the bioactivity of the interface is compromised (see e.g. our work in *Acta Biomater* 77, 74 (2018) <https://doi.org/10.1016/j.actbio.2018.07.016>).

This phenomenon highlights the importance of the initial cell-material interactions and how this determines cell response in the mid and long term. The reviewer made the comment of whether these holes in the substrate can be afterwards occupied by, e.g., proteins coming from the media. We would point out that cells very quickly start producing their own ECM (see e.g. *Nat Mater* 18, 883, 2019 <https://doi.org/10.1038/s41563-019-0307-6>) and this does not prevent the initial effect from the substrate. We therefore argue that, while in the mid/long terms some reorganization of the bilayers occurs, cellular responses are still governed by the initial

interactions and hence by the varying degree of viscosity of the substrates. Indeed, if cell response was driven by the effect of these defects, we would then expect a similar response to all surfaces, corresponding to the phenotype of cells seeded directly on glass. Instead, we do observe clear differences in cell behaviour depending on the substrate, here and in our previous work (Bennett et al., PNAS 2018; <https://doi.org/10.1073/pnas.1710653115>).

Following the reviewer's advice we included additional text in the revised version R2 of the manuscript to address this comment.

2) Mechanism of cross-talk.

The anti-N-cadherin experiments are interesting and much appreciated. However, Supplementary Fig. 13a shows that the anti-N-cadherin application does increase actin flow compared to RGD alone, particularly for DPPC. This could be due to the fact that the antibody could also be reaching the ventral side of the cells, as it is applied in solution. Regardless, addressing this change in actin flow is needed, whether through more extensive discussion on this mechanism on the overall impact of this study or additional experiments that more completely separate the signals. These experiments could include exposure of cells to beads coated with the anti-N-cadherin antibodies or using micropatterned surfaces with small regions of anti-N-cadherin interspersed into the lipid bilayer.

We thank the reviewer for pointing out that we did not discuss the increase in actin flow following treatment with anti-N-cadherin on RGD-functionalised DPPC. We acknowledge that a small effect is observed, whereby the actin flow increases after N-cadherin application on RGD-DPPC as seen in Supplementary Figure 13a. However, this effect is smaller than the one observed when ventral HAVDI ligation occurs, and, importantly, is not accompanied by a significant change in focal adhesion formation, as seen in Supplementary Figure 11b. We therefore argue that the local competition for actin fibres between integrins and cadherins remains the key mechanism to regulate cross-talk following ventral HAVDI engagement. As the reviewer suggests, the effect that we see on the actin flow may be due to some of the antibody reaching the ventral side of the cells and eliciting this subtle change. We have improved our discussion of these results in the revised version R2 of the manuscript to point out these changes.

Reviewer #3:

I would like to extend my congratulations to the authors for their heroic effort in addressing all of my comments and remarks. All of my questions have been successfully addressed, and I believe this work is now fit for publication in its current state.

We are delighted that the reviewer appreciates the significant amount of work that we put into R1 to address their concerns.

1. I have some questions regarding the concentration of peptides used, and what they mean. In the methods section the authors wrote that biotinylated lipids were used in different quantities to achieve 0.02, 0.2, 0.22, 2 or 2.2 % of functionalization (correct?). Is this the limiting factor on the functionalization of the bilayers? For instance, for a bilayer with 0.2% RGD + 2% HAVDI, a lipid mixture containing 2.2% of biotinylated lipids is used and a solution of 1:10 of RGD:HAVDI is added to the solution? Can you please clarify this. Since the biotinylated lipids are always DPPC, will there be an effect on the viscosity between the layers with 0.02 and the 2.2% (for the DOPC samples)?
2. The authors state that they use 0.2% and 2% HAVDI concentration as the 'low' and 'high' HADVI. In supporting figure 4, 10% HAVDI was used. Can the authors provide some rationale to why the 0.2 and 2% HAVDI conditions were chosen? And how was the layer with 10% HAVDI prepared?
3. For the quantification of the peptides in supplementary figure 3 I have a couple of comments. First, it would be useful to have a control with the intensities of the bilayers containing only streptavidin. Then we could also confirm the presence of RGD and HADVI at the lower concentration. Secondly, the image presented on panel seems brighter, while the quantification shows that there is no difference. Is this because of the LUT of the image or was this not a very representative image? Also, the amount of HAVDI is 10x higher but the fluorescence intensity detected is only 2-2.5x higher – any explanations? And, finally, since there are a lot of experiments presented in the paper with varying concentrations of RGD, I would also quantify the amount of RGD only in the layers, to be sure that there is an increase.
4. What do the author mean by 'a higher intensity of N-cadherin expression' (end page 5). The supplementary figure 3 shows the quantification of the antibody in panel a, but the raw images are not shown. How did the authors quantify this? Is this per cell, an average? Just a cross-section or the whole volume? Depending on how the images are analyzed, the difference can be related to the different cellular localization instead of an expression level.
5. "when lower concentrations (0.02 % mol) of RGD were tested (i.e. higher RGD spacing), changes in the size of hMSC area due to HAVDI were observed only for DPPC and not for DOPC" – it is not clear why are there differences between DOPC and DPPC in this case; please add some explanation.
6. The data presented in figure 1e,f,g and in supplementary figure 4b is partially referent to the same experimental conditions but the trends shown are different. For instance, DPPC with 0.2%RGD, the cell area in supplementary figure 4 increases when going from 0.02 to 0.2% HAVDI, which is not in agreement with the trend discussed in the main text.
7. Regarding the colocalization of MIIA with the focal adhesions, I think that the data does not support the conclusions in the manuscript. The authors stated "adding HAVDI decreases MIIA levels int eh focal adhesions". By looking at the images, it seems that the presence of HAVDI leads to a much higher expression of MIIA (brighter cells). It is difficult to see if there is MIIA present at the focal adhesions because the level in the cytosol is much higher. The authors show intensity profiles, but it is difficult to interpret these without knowing their location of the image. To retrieve some quantitative analysis, the authors could do some image correlation analysis, rather than showing one intensity profile for each condition.
8. On figure 2, the data shown in panel b, and on panels c and d is different, while it is acquired for the experimental conditions (e.g. compare DPPC RGD only in panel b and

panel d). Can the authors explain this? This is especially important since in panel d there is no decrease in the nuc/cyt ratio of YAP between RGD only and low HAVDI, while the authors claim a difference in panel b (with $p < 0.0001$)...

9. For the differentiation markers (figure 1g, h, i), to be able to draw conclusion on the effect of viscosity, it is necessary to calculate/show the statistical differences between the same condition in different substrates. For a first look, it seems that the data is too spread to be able to draw any conclusions (differences not statistically significant). This is valid for all the expression markers analysed.
10. "Interestingly, at increasing concentrations of HAVDI, SOX9 expression increased independently of viscosity." Except for the glass, where there seems to be a drop. Can you comment on this?
11. The results regarding the intensity and length of the focal adhesion are difficult to interpret due to the lack of information regarding the analysis. How is the intensity calculated? Is there any image segmentation performed? How?
12. On p14 the authors claim "only for more viscous substrates, smaller FAs when adding HAVDI". The differences between RGD, low HADVI, high HAVDI should be quantified in all viscosity conditions (also graph 3e).
13. For Vinculin, only the quantification is shown (Figure 3e). Please add the fluorescence images as well. Add also the images, for FAK and vinculin, for low HAVDI (even if just in SI).
14. The authors claim that "by overexpressing talin, FAs increase." They should calculate statistical significances between each condition for non-transfected and transfected cells.
15. The authors claim that "as expected and in contrast to what we observed in control cells on DPPC, the length of FAs in transfected Y201 cells was not affected by HAVDI" and "as expected and in contrast to what we observed in control cells (= no transfection I assume) on DPPC AND GLASS (?), the length of FAs in transfected Y201 cells was not affected by HAVDI". The authors should show statistical differences between RGD-low-high in transfected cells in all conditions.
16. "In this model, talin-vinculin and α -catenin-vinculin would compete with each other to bind to actin, given the limited availability of actin filaments". Could this also be related to the availability of vinculin?
17. "When low-viscosity bilayers (DOPC) are functionalized with HAVDI, this functionalization does not affect FA formation". Can the authors provide some hypothesis as to why this is happening?
18. "hTERT Y201 MSCs behave in the same way as primary hMSCs, being mechanosensitive and with a similar differentiation potential (Supplementary Figure 9)". I disagree with this statement. If the authors want to claim this, they should put both cell lines next to each other in the same graphs. Supplementary figure 9 is only about Y201 cells. For example SOX9 levels seem quite different in both cell lines: primary cells have SOX9 0.05-0.15 (fig 2h) while Y201 have 2-4 (fig S9h). Also, the trend between the RGD-low-high for DPPC and glass are different between the 2 cell lines. And the cyt/nuc ratio for YAP is very different, suggesting that the cells ARE responding differently.
19. "We observed that increased talin expression led to more vinculin being recruited to the site of FAs". So maybe vinculin is drawn away from N-cadherin adhesions, and this is the rate-limiting step (not actin)? This claim could be supported by imaging N-cadherin with talin overexpression to check if there was less cadherin present.

20. Add more information on the quantification and statistical analysis. For instance, in many cases, the total number of observations is not mentioned, nor the number of biological replicates.
21. % mol, mol %, % are used interchangeably, make more uniform throughout the manuscript
22. Please add also the lower error bars in the bar graphs (in both the manuscript and the SI)
23. The letters and text describing the viscosity values in figure 1a are too small to read. Can you also mention the source of these values or how were they obtained?
24. Figure 1c, there seems to be no data for the non-functionalized surface (no error bar visible)
25. For figure 1h, it would be better to have an overview image (similar to that shown in sup. Fig. 3b). If you opt for showing the individual cells, please center the images...
26. Scale bars: either always in the figures or in the caption (eg. fig 1b versus 1h). Also, please use μm instead of um.
27. Mention supplementary figure 5 in the main text.
28. Add the time scales for the kymographs in figure 4.
29. "at a density of 10000 cells/cm² for fixed cell experiments and 20000 cells/cm² for in-cell westerns assays." Why the different densities? Does this have an impact on the results?
30. Please review the methods section of typos (there are a LOT).

Day 1

DPPC